# EFFICIENT SALIENCY MAPS FOR EXPLAINABLE AI

## ABSTRACT

We describe an explainable AI saliency map method for use with deep convolutional neural networks (CNN) that is much more efficient than popular gradient methods. It is also quantitatively similar and better in accuracy. Our technique works by measuring information at the end of each network scale which is then combined into a single saliency map. We describe how saliency measures can be made more efficient by exploiting Saliency Map Order Equivalence. Finally, we visualize individual scale/layer contributions by using a Layer Ordered Visualization of Information. This provides an interesting comparison of scale information contributions within the network not provided by other saliency map methods. Since our method only requires a single forward pass through a few of the layers in a network, it is at least 97x faster than Guided Backprop and much more accurate. Using our method instead of Guided Backprop, class activation methods such as Grad-CAM, Grad-CAM++ and Smooth Grad-CAM++ will run several orders of magnitude faster, have a significantly smaller memory footprint and be more accurate. This will make such methods feasible on resource limited platforms such as robots, cell phones and low cost industrial devices. This will also significantly help them work in extremely data intensive applications such as satellite image processing. All without sacrificing accuracy. Our method is generally straight forward and should be applicable to the most commonly used CNNs [1].

## 1 INTRODUCTION

Deep neural networks (DNN) have provided a new burst of research in the machine learning community. However, their complexity obfuscates the underlying processes that drive their inferences. This has lead to a new field of *explainable AI* (XAI). A variety of tools are being developed to enable researchers to peer into the inner workings of DNNs. One such tool is the XAI saliency map. It is generally used with image or video processing applications and is supposed to show what parts of an image or video frame are most important to a network's decisions. The seemingly most popular methods derive a *gradient* saliency map by back-propagating a gradient from the end of the network and project it onto an image plane (Simonyan et al., 2014; Zeiler & Fergus, 2014; Springenberg et al., 2015; Sundararajan et al., 2017; Patro et al., 2019). The gradient can typically be from a loss function, layer activation or class activation. Thus, it requires storage of the data necessary to compute a full backward pass on the input image.

Several newer methods attempt to iteratively augment the image or a mask in ways that affect the precision of the results (Fong & Vedaldi, 2017; Chang et al., 2018). Additionally, saliency map encoders can be trained within the network itself (Dabkowski & Gal, 2017). Both of these methods have a distinct advantage of being more self-evidently empirical when compared with gradient techniques. *Class Activation Map* (CAM) methods (Selvaraju et al., 2017; Chattopadhyay et al., 2018; Omeiza et al., 2019) efficiently map a specified class to a region in an image, but the saliency map is very coarse. They generally use a method like *Guided Backprop* (Springenberg et al., 2015) to add finer pixel level details. This requires a full backwards pass through the network, and it adds significant memory and computational overhead to CAM solutions relative to just computing the CAM alone. Several of the CAM methods compute gradients aside from the use of Guided Backprop, but we will differentiate them by referring to them as CAM methods.

---

[1]For reproducibility, full source code is available at http://www.anonymous.submission.com

## 1.1 EFFICIENCY AND WHY IT MATTERS HERE

Most saliency map methods require many passes through the network in order to generate results or train. The gradient methods hypothetically would require just one backwards pass, but often require as many as 15 in order to give an accurate rendering (Hooker et al., 2018). This is not always a problem in the lab when a person has a powerful GPU development box. However, what if one would like to see the workings of the network at training time or even run time? This can be a serious hurdle when running a network on mobile or embedded platforms. It is not uncommon for hardware to be barely fast enough to process each image from a video source. Running a full backward pass can lead to dropped frames. Additionally, viewing or saving the saliency map for *each* frame is infeasible. Another problem is that some platforms may not have the memory capacity to save all the information required for a backward pass. Gradient based methods cannot work in these instances. Sometimes this can even be the case with powerful hardware. Satellite images can be vary large and potentially exhaust generous resources. An efficient method would enable developers in these areas to get feedback in the field and aid in debugging or understanding the behavior of a network.

Here we show a method of computing an XAI saliency map which is highly efficient. The memory and processing overhead is several orders of magnitude lower than the commonly used gradient methods. This makes it feasible in any resource limited environment. Also, since we demonstrate empirically that our method is either similar or more accurate than the most commonly used gradient methods, it can be used it to speed up run-time in any situation. It is fast enough that we already use it automatically when training networks. We notice very little degradation of training speed.

## 2 METHODS

### 2.1 SALIENCY MAP DERIVATION

We were looking for a method to compute saliency maps based on certain conditions and assumptions.

1. The method used needs to be relatively efficient to support rapid analysis at both test time and during DNN training.

2. The method should have a reasonable information representation. As a DNN processes data, the flow of information should become localized to areas which are truly important.

3. The method should capture the intuition that the informativeness of a location is proportional to the overall activation level of all the filters as well their variance. That is, informative activations should have a sparse pattern with strong peaks.

Our approach works by creating saliency maps for the output layer of each *scale* in a neural network and then combines them. We can understand scale by noting that the most commonly used image processing DNNs work on images with filter groups at the same scale which down-sample the image and pass it to the group of filters at the next scale, and so on. Given a network like ResNet-50 (He et al., 2015) with in input image size of 224$x$224, we would have five scale groups of size 112$x$112, 56$x$56, 28$x$28, 14$x$14 and 7$x$7. It is at the end of these scale groups where we are interested in computing saliency. In this way, our approach is efficient and is computed during the standard forward pass through the network. No extra pass is needed.

To achieve localization of information, we measure statistics of activation values arising at different input locations. Given an output activation tensor $\mathbf{T} \in \mathbb{R}^{+, p \times q \times r}$ with spatial indices $i, j \in p, q$ and depth index $k \in r$ from some layer $\mathbf{T} = l(\mathbf{X})$. In our case $l(.)$ is a *ReLU* (Nair & Hinton, 2010). We apply a function to each column at $i, j$ over all depths $k$. This yields a 2D saliency map $\boldsymbol{S} \in \mathbb{R}^{+, p \times q}$ where $\boldsymbol{S} = \varphi(\mathbf{T})$. We process the tensor after it has been batch-normalized (Ioffe & Szegedy, 2015) and processed by the activation function. When we compute Truncated Normal statistics as an alternative in later section, we take the tensor prior to the activation function.

Finally, to capture our intuition about the informativeness of an output activation tensor, we derived $\varphi(.)$ by creating a special simplification of the maximum likelihood estimation *Gamma Scale* parameter (Choi & Wette, 1969). One way we can express it is:

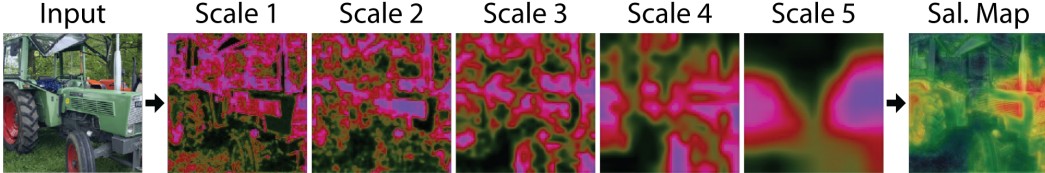

Figure 1: The left most image is the input to the network. Five saliency maps are shown for each spatial scale in the network. They are combined per Eq 3. The right most image is the combined saliency map created from these. To aid in visualizing context, it has been alpha blended with a gray scale version of the original image here at 25%. Many more combined saliency map examples can be seen in *Appendix* Figures 9 and 10

.

$$\varphi\left(.\right) = \frac{1}{r} \cdot \sum_{k=1}^{r} x_k \cdot \left[ log_2 \left( \frac{1}{r} \cdot \sum_{k=1}^{r} x_k \right) - \frac{1}{r} \cdot \sum_{k=1}^{r} log_2 \left( x_k \right) \right] \tag{1}$$

To avoid log zero, we add $1e - 06$ to each $x_k$. How mean and variance relate seems readily apparent with the square bracketed part being the computational formula for the standard deviation with values taken to $log_2\left(.\right)$ rather than squared. This is preceded by a typical mean estimate. This meets the third requirement we mentioned. This simplification is **Saliency Map Order Equivalent (SMOE)** to the full iterative (and expensive) scale parameter estimation. We define SMOE as follows. Given saliency map $S_a \in \mathbb{R}^{+,p \times q}$ and $S_b \in \mathbb{R}^{+,p \times q}$ where we may have $S_a \neq S_b$, if we sort the pixels by value, then $S_a$ will be sorted in exactly the same order as $S_b$. That means that the most salient location $i, j$ is exactly the same in both $S_a$ and $S_b$. This also means that if we create a binary mask of the $n\%$ most salient pixels, the mask for both $S_a$ and $S_b$ will also be exactly the same. SMOE is preserved if for instance, we apply independent monotonic functions to a saliency map. As such, we may as well strip these away to save on computation. Tie values may create an imperfect equivalence, but we assert that these should be very rare and not affect results by a measurable amount.

Using $\mu$ as the mean of each column $r$ in **T**, we can see the information relation more clearly if we simplify Eq 1 further which gives us our **SMOE Scale** method:

$$\varphi\left(.\right) = \frac{1}{r} \cdot \sum_{k=1}^{r} \mu \cdot \log_2 \left( \frac{\mu}{x_k} \right) \tag{2}$$

The resemblance to conditional entropy is apparent. However, since the values in Eq 2 are not probabilities, this does not fit the precise definition of it. On the other hand, the interpretation is fairly similar. It is the mean activation multiplied by the information we would gain if we knew the individual values which formed the mean. Put in traditional terms, it is the information in the mean conditioned on the individual values. Numerical examples of this method at work can be seen in *Appendix* Table 3 along with more information on the derivation. To create a 2D saliency map $S \in \mathbb{R}^{+,p \times q}$, we simply apply Eq 2 at each spatial location $i, j \in p, q$ with column elements $k \in r$ in the 3D activation tensor **T** for a given input image.

## 2.2 COMBINED SALIENCY MAP GENERATION

For each input image, we derive five saliency maps. For different networks, this number may vary. Given a network such as a ResNet (He et al., 2015), AlexNet (Krizhevsky et al., 2013), VGG Net (Simonyan & Zisserman, 2015) or DenseNet (Huang et al., 2017) we compute saliency on the final tensor computed at each spatial scale. Recall that most image classification networks process images in a pipeline that processes an image in consecutive groups of convolution layers where each group downsamples the image by 1/2x before passing it onto the next. It is just prior to the downsampling that we compute each saliency map. Computing saliency across image scales is a classical technique (Itti et al., 1998). This is also similar to the method used in the XAI saliency technique described in (Dabkowski & Gal, 2017).

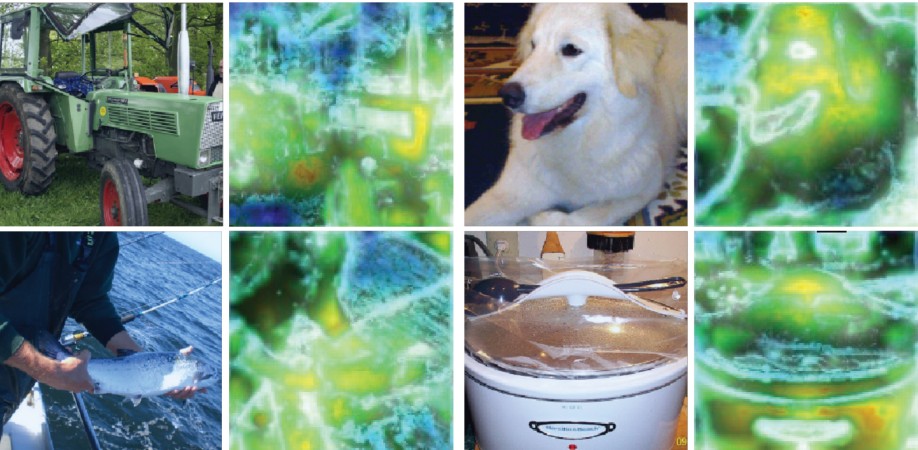

Figure 2: Images are shown with their combined saliency map using our *LOVI* scheme. The hue in each saliency map corresponds to layer activation. Earlier layers start at violet and trend red in the last layers following the order of the rainbow. Areas which are blue or violet are only activated early in network processing. They tend to activate early filters, but are later disregarded by the network. Yellow and red areas are only activated later in the network. They appear to be places where the objects components are combined together. White areas are activated throughout all network layers. They possibly correspond to the most important features in the image. Many more examples can be seen in *Appendix* Figures 11 and 12

.

To make our maps easier to visualize or combine together, we normalize them from 0 to 1 by squashing them with the normal *cumulative* distribution function $\gamma(s; \mu, \sigma)$. Here mean and standard deviation are computed independently over each saliency map. We then create a combined saliency map by taking the weighted average of the maps. Since they are at different scales, they are upsampled via bilinear interpolation to match the dimensions of the input image. Given $r$ saliency maps that have been bilinear interpolated (upsampled) to the original input image size $p, q$, they are then combined as:

$$c_{i,j} = \frac{\sum_{k=1}^{r} \gamma(s_{i,j,k}; \mu_k, \sigma_k) \cdot w_k}{\sum_{k=1}^{r} w_k} \tag{3}$$

Note that technically, we compute $\gamma(s; \mu, \sigma)$ *before* we upsample. Weighting is very useful since we expect that saliency maps computed later in the network should be more accurate than saliency maps computed earlier as the network has reduced more irrelevant information in deeper layers, distilling relevant pixels from noise (Tishby et al., 2000). Network activity should be more focused on relevant locations as information becomes more related to the message. We observe this behavior which can be seen later in Figure 4. A saliency map generation example can be seen in Figure 1 with many more examples in *Appendix* Figures 9 and 10.

The advantages of creating saliency maps this way when compared with most gradient methods are:

- **Pro**: This is relatively efficient, requiring processing of just five low cost layers during a standard forward pass.
- **Pro**: We can easily visualize the network at different stages (layers).
- **Con**: The current method does not have a class specific activation map (CAM) (Selvaraju et al., 2017; Chattopadhyay et al., 2018; Omeiza et al., 2019), but we discuss how this can be done later.

## 2.3 VISUALIZING MULTIPLE SALIENCY MAPS

One advantage to computing multiple saliency maps at each scale is that we can get an idea of what is happening in the middle of the network. However, with so many saliency maps, we are starting to be overloaded with information. This could get even worse if we decided to insert saliency maps after each layer rather than just at the end of each scale. One way to deal with this is to come up with

a method of combining saliency maps into a single image that preserves useful information about each map. Such a composite saliency map should communicate where the network is most active as well as which layers specifically are active. We call our method **Layer Ordered Visualization of Information (LOVI)**. We do this by combining saliency maps using an *HSV* color scheme (Joblove & Greenberg, 1978) where *hue* corresponds to which layer is most active at a given location. That is, it shows the mean layer around which a pixels activity is centered. *Saturation* tells us the uniqueness of the activation. This is the difference between the maximum value at a location and the others. *Value* (intensity) corresponds to the maximal activation at that location. Basically, this is a pixel's importance.

If only one layer is active at a location, the color will be very saturated (vivid colors). On the other hand, if all layers are equally active at a given location, the pixel value will be unsaturated (white or gray). If most of the activation is towards the start of a network, a pixel will be violet or blue. If the activation mostly happens at the end, a pixel will be yellow or red. Green indicates a middle layer. Thus, the color ordering by layer follows the standard order of the rainbow. Examples can be seen in Figure 2. Given $k \in r$ saliency maps $\boldsymbol{S}$ (in this instance, we have $r = 5$ maps), we stack them into a tensor $\mathbf{S} \in \mathbb{R}^{+, p \times q \times r}$. Note that all $s \in [0, 1]$ because of Eq 3 and they have been upsampled via bilinear interpolation to match the original input image size. Given:

$$\phi\left(k\right) = 1 - \frac{k-1}{r} \cdot \left(\frac{r-1}{r}\right)^{-1}, \nu = \frac{1}{r} \tag{4}$$

**Hue** $\in [0, 360]$ is basically the center of mass of activation for column vector $\boldsymbol{s}$ at each location $i, j \in p, q$ in $\mathbf{S}$:

$$Hue = \frac{\sum_{k=1}^{r} s_k \cdot \phi\left(k\right)}{\sum_{k=1}^{r} s_k} \cdot 300 \tag{5}$$

**Saturation** $\in [0, 1]$ is the inverse of the ratio of the values in $\boldsymbol{s}$ compared to if they are all equal to the maximum value. So for instance, if one value is large and all the other values are small, saturation is high. On the other hand, if all values are about the same (equal to the maximum value), saturation is very small:

$$Sat = 1 - \frac{\sum_{k=1}^{r}\left(s_k\right) - \nu}{r \cdot max\left(\boldsymbol{s}\right) \cdot \left(1 - \nu\right)} \tag{6}$$

**Value** $\in [0, 1]$ is basically just the maximum value in vector $\boldsymbol{s}$ :

$$Val = max\left(\boldsymbol{s}\right) \tag{7}$$

Once we have the HSV values for each location, we then convert the image to RGB color space in the usual manner.

## 2.4 Quantification Via ROAR and KAR

(Hooker et al., 2018) proposed a standardized method for comparing XAI saliency maps. This extends on ideas proposed by (Dabkowski & Gal, 2017; Samek et al., 2017) and in general hearkens back to methods used to compare computational saliency maps to psychophysical observations (Itti & Koch, 2001). The general idea is that if a saliency map is an accurate representation of what is important in an image, then if we block out salient regions, network performance should degrade. Conversely, if we block out non-salient regions, we should see little degradation in performance. The ROAR/KAR metrics measure these degradations explicitly. The KAR metric (Keep And Retrain) works by blanking out the *least salient* information/pixels in an input image, and the ROAR (Remove And Retrain) metric uses the contrary strategy and removes the *most salient* pixels. Figure 3 shows an example of ROAR and KAR masked image. A key component to the ROAR/KAR method is that the network needs to be retrained with saliency masking in place. This is because when we mask out regions in an input image, we unavoidably create artifacts. By retraining the network on masked images, the network learns to ignore the new artifacts and focus on image information.

We will give a few examples to show why we need both metrics. If a saliency map is good at deciding which parts of an image are least informative but gets the ranking of the most salient objects wrong, ROAR scores will suggest the method is very good. This is because it masks out the most salient locations in one large grouping. However, ROAR will be unable to diagnose that the saliency map has erroneously ranked the most informative locations until we have removed 50% or more of the salient pixels. As such, we get no differentiation between the top 1% and the top 10% most salient pixels. On the other hand, KAR directly measures how well the saliency map has ranked the most informative locations. By using both metrics, we can quantify the goodness of both the most and least salient locations in a map.

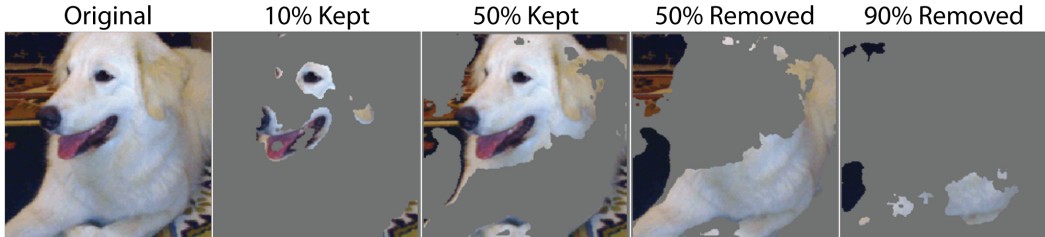

Figure 3: These are the KAR (kept) and ROAR (removed) mask images created by masking out the original images with the combined saliency map. The percentage is how much of the image has been kept or removed based on the combined saliency map. Thus, the *10%* kept example shows the top 10% most salient pixels in the image. It is these example images that are fed into the network when we compute the KAR and ROAR scores. Many more examples can be seen in *Appendix* Figure 8 .

# 3 QUANTITATIVE EXPERIMENTS

## 3.1 COMPARING DIFFERENT EFFICIENT STATISTICS

We tested our SMOE Scale saliency map method against several other statistical measures using three different datasets that have fairly different tasks and can be effectively trained from scratch. The sets used are ImageNet (Deng et al., 2009), CSAIL Places (Zhou et al., 2014) and COWC (Mundhenk et al., 2016). ImageNet as a task focuses on foreground identification of objects in standard photographic images. Places has more emphasis on background objects, so we would expect more spatial distribution of useful information. COWC, *Cars Overhead With Context* is an overhead dataset for counting as many as 64 cars per image. We might expect information to be spatially and discretely localized, but distributed over many locations. In summary, these three datasets are expected to have fairly different distributions of important information within each image. This should give us more insight into performance than if we used several task-similar datasets (e.g. Three photographic foreground object sets such as; ImageNet + CUB (birds) (Welinder et al., 2010) + CompCars (Yang et al., 2015)).

For compatibility with (Hooker et al., 2018), we used a ResNet-50 network (He et al., 2015). We also show performance on a per layer basis in order to understand the accuracy at different levels of the network. For comparison with our SMOE Scale method, we included any statistical measure which had at least a modicum of justification and was within the realm of the efficiency we were aiming for. These included parameter and entropy estimations from *Normal, Truncated-normal, Log-normal* and *Gamma Distribution* models. We also tested *Shanon Entropy* and *Renyi Entropy*. To save compute time, we did a preliminary test on each method and did not continue using it if the results qualitatively appeared very poor and highly unlikely to yield good quantitative results. Normal entropy was excluded because it is SMOE with the Normal standard deviation. This left us with nine possible statistical models which we will discuss in further detail.

Saliency maps for each method are computed over each tensor column in the same way as we did with our SMOE Scale metric. The only difference is with the truncated-normal statistic which computes parameters prior to the ReLU layer. We should note that (Jeong & Shin, 2019) uses a truncated normal distribution to measure network information for network design. Recall that we have five saliency map layers. They are at the end of each of the five network spatial scales. We test each one at a time. This is done by setting the network with pre-trained weights for the specific dataset. Then, all weights in the network which come after the saliency mask to be tested are allowed to fine-tune over 25 epochs. Otherwise, we used the same methodology as (He et al., 2015) for data augmentation etc. This is necessary in order to adapt the network to mask related artifacts as specified in the ROAR/KAR protocol. At the level where the saliency map is generated, we mask out pixels in the activation tensor by setting them to zero. For this experiment, we computed the ROAR statistic for the 10% least salient pixels. For KAR, we computed a map to only let through the top 2.5% most salient pixels. This creates a more similar magnitude between ROAR and KAR measures. Layer scores for the top five methods can be seen in Figure 4. We combine layer scores two different ways since ROAR and KAR scores are not quite proportional. These methods both yield very similar results. The *first method* takes the improvement difference between

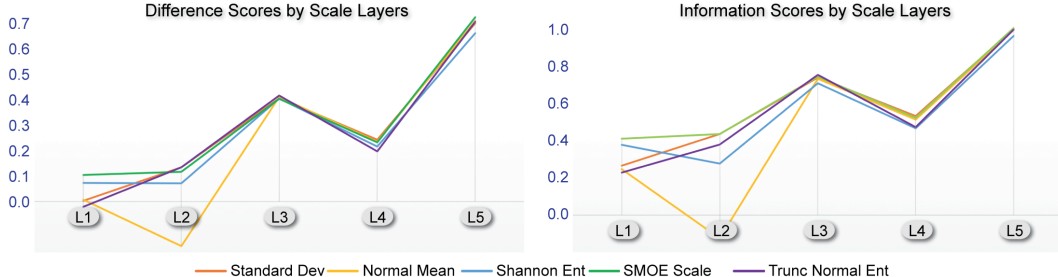

Figure 4: SMOE Scale is compared with several other efficient statistical methods. The **Y-axis** is the **combined score per scale layer** over all three image sets. The **X-axis** is the **network layer** with L1 being the earliest layer in the network and L5 near the end. A difference score of *zero* means the result was about the same as for a randomly generated saliency map. Less than zero means it is worse. SMOE Scale differentiates itself the most early on in the network where most statistical methods score at the level of a random saliency map. About mid way through, the difference between methods becomes relatively small. This may be because information contains more message and less noise by this point in processing. Finer grain details can be seen in *Appendix* Table 4
.

Table 1: **KAR and ROAR results per dataset**. The *Difference Score* shows the results using Eq 8. The *Information Score* uses Eq 9. They are sorted by the average difference score (AVG). The SMOE Scale from Eq 2 performs best overall using both scoring methods. The vanilla standard deviation is second best. Recall it is SMOE with normal entropy. Truncated normal entropy is best on the COWC set and ranks third overall. It is interesting to note that the difference in scores over COWC are not as large as the other two datasets. The top four methods all are information related and mean activation style methods are towards the bottom. Finer grain details can be seen in *Appendix* Table 4
.

| Method | **Difference Score** | | | | **Information Score** | | | |
| --- | --- | --- | --- | --- | --- | --- | --- | --- |
| | ImNet | Places | COWC | AVG | ImNet | Places | COWC | AVG |
| SMOE Scale | **1.70** | **0.90** | 1.61 | **1.40** | **1.13** | **0.68** | 1.31 | **1.04** |
| Standard Dev | 1.64 | 0.83 | 1.61 | 1.36 | 1.07 | 0.61 | 1.30 | 0.99 |
| Trunc Normal Ent | 1.56 | 0.77 | **1.64** | 1.32 | 1.00 | 0.56 | **1.32** | 0.96 |
| Shanon Ent | 1.61 | 0.80 | 1.51 | 1.31 | 0.98 | 0.59 | 1.23 | 0.93 |
| Trunc Normal Std | 1.51 | 0.71 | **1.64** | 1.28 | 1.00 | 0.52 | **1.32** | 0.94 |
| Trunc Normal Mean | 1.38 | 0.67 | **1.64** | 1.23 | 0.96 | 0.49 | **1.32** | 0.92 |
| Normal Mean | 1.29 | 0.63 | 1.42 | 1.11 | 0.75 | 0.44 | 1.18 | 0.79 |
| Log Normal Ent | 1.16 | 0.66 | 1.44 | 1.09 | 0.82 | 0.47 | 1.20 | 0.83 |
| Log Normal Mean | 1.46 | 0.55 | 1.09 | 1.03 | 0.54 | 0.35 | 0.88 | 0.59 |

tested method's score and a randomized mask score. We have five $\kappa \in [0, 1]$ KAR scores for a method. We have five $\rho \in [0, 1]$ ROAR scores for a method and five $z \in [0, 1]$ scores from a random mask condition. This corresponds to each saliency map spatial scale which we tested. We compute a simple difference score as:

$$D(\rho, \kappa) = \sum_{p=1}^{5} (z_p - \rho_p) + \sum_{q=1}^{5} (\kappa_q - z_q) \tag{8}$$

The *second method* is an information gain score given by:

$$I(\rho, \kappa) = -\sum_{p=1}^{5} \rho_p \cdot log_2\left(\frac{\rho_p}{z_p}\right) - \sum_{q=1}^{5} z_q \cdot log_2\left(\frac{z_q}{\kappa_q}\right) \tag{9}$$

Table 1 shows the results.

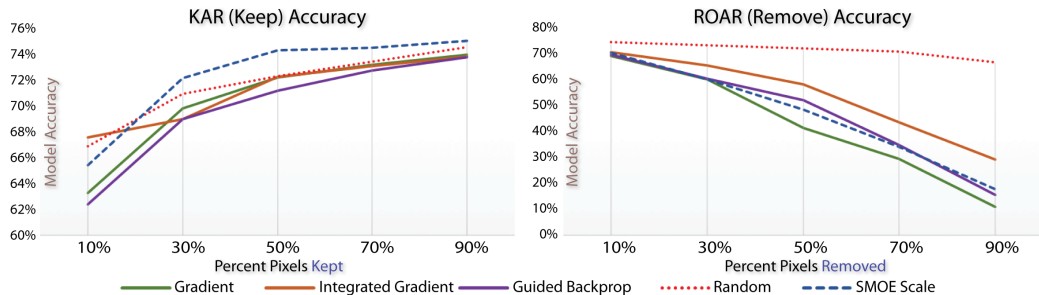

Figure 5: SMOE Scale with prior layer weights is compared with three popular baseline methods that all use Squared SmoothGrad. Scores for these three are taken from (Hooker et al., 2018). The **Y-axis** is the **model accuracy** on *ImageNet* only. The **X-axis** is how much of the input image **salient pixels are kept or removed**. KAR keeps the most salient locations. Higher accuracy values are better. ROAR removes the most salient regions. Lower values are better. Our method does not seem to suffer as much when the 10% least salient parts are removed in KAR and in general maintains a better score. Our ROAR scores are very similar to Guided Backprop. Finer grain details can be seen in *Appendix* Table 5. Note that these results are *not* per layer. For a closer numerical comparison with the mask layer method, see *Appendix* Table 4.

## 3.2 COMPARISON WITH POPULAR METHODS

We compare our method with three popular saliency map methods using the standard ROAR/KAR methodology. These are *Gradient Heatmaps* (Simonyan et al., 2014), *Guided Backprop* (Springenberg et al., 2015) and *Integrated Gradients* (Sundararajan et al., 2017). All methods use *SmoothGrad-Squared* (Smilkov et al., 2017) which gives generally the best results. We should note that without SmoothGrad or another augmentation, all three do not yield very good ROAR/KAR scores.

We compare three different weighting strategies when combining the saliency maps from all five scales. In the first strategy, we weight all five maps equal [1,1,1,1,1]. In the second, we use a rule-of-thumb like approach where we weight the first layer the least since it should be less accurate. Then each successive layer is weighted more. For this we choose the weights to be [1,2,3,4,5]. The third method weights the maps based on the expected accuracy given our results when we computed Table 1. These prior weights are [0.18, 0.15, 0.37, 0.4, 0.72]. The reason for showing the rule-of-thumb results is to give an idea of performance given imperfect weights since one may not want to spend time computing optimal prior weights.

To fairly compare the three popular saliency map methods with our own, we adopt a methodology as close as possible to (Hooker et al., 2018). We train a ResNet-50 from scratch on ImageNet with either ROAR or KAR masking (computed by each of the different saliency mapping approaches in turn) at the start of the network. Otherwise, our training method is the same as (He et al., 2015). The comparison results are shown in Figure 5. We can try and refine these results into fewer scores by subtracting the sum of the ROAR scores from the sum of the KAR scores. The results can be seen in Table 2. The KAR score for our method is superior to all three comparison methods. The ROAR score is better than Guided Backpropagation and Integrated Gradients. This suggests our method is superior at correctly determining which locations are most salient, but is not as good as Gradient Heatmaps for determining which parts of the image are least informative.

## 4 DISCUSSION

### 4.1 METHOD EFFICIENCY

The method as proposed is much faster than the three baseline comparison methods. Given a ResNet-50 network, we only process five layers. The other methods require a special back propagation step over all layers. We can compute the cost in time by looking at operations that come from three sources. The first is the computation of statistics on tensors in five layers. The second is

Table 2: **Combined KAR and ROAR scores for several methods on ImageNet Only**. The top three rows show several popular methods with *Squared SmoothGrad* (Smilkov et al., 2017). These scores are created by simply summing the individual scores together. ROAR is negative since we want it to be as small as possible. *Prior Layer Weights* means we applied layer weights based on the prior determined accuracy of the layer saliency map. We include our top three scoring methods. The SMOE Scale method outperforms the three baseline methods on KAR. It outperforms Guided Back-prop and Integrated Gradients on ROAR as well as overall. The Gradient method is best overall, but as we discuss later, it is at least *1456 times* more computationally expensive to compute. Truncated normal entropy scores about the same as SMOE Scale. Since SMOE Scale gains its largest performance boost in the earlier layers, when we apply prior weighting, we reduce that advantage. Finer grain details can be seen in *Appendix* Table 5.

| Method | KAR | ROAR | SUM |
|---|---|---|---|
| Gradient (Simonyan et al., 2014) | 3.57 | -3.54 | 0.04 |
| Guided Backprop (Springenberg et al., 2015) | 3.60 | -3.57 | 0.04 |
| Integrated Grad (Sundararajan et al., 2017) | **3.62** | -3.58 | 0.03 |
| Gradient -*w*- SmoothGrad Sq. | 3.52 | **-2.12** | **1.41** |
| Guided Backprop -*w*- SmoothGrad Sq. | 3.49 | -2.33 | 1.16 |
| Integrated Grad -*w*- SmoothGrad Sq. | 3.56 | -2.68 | 0.88 |
| SMOE Scale + Prior Layer Weights | 3.61 | **-2.31** | **1.30** |
| SMOE Scale + Layer Weights [1,2,3,4,5] | **3.62** | -2.34 | 1.28 |
| SMOE Scale + Layer Weights [1,1,1,1,1] | **3.62** | -2.46 | 1.15 |
| Normal Std + Prior Layer Weights | 3.61 | -2.32 | 1.29 |
| Trunc Normal Ent + Prior Layer Weights | 3.61 | **-2.31** | **1.30** |

the normalization of each 2D saliency map. Third we account for the cost of combing the saliency maps.

Ops for our solution ranges from $1.1x10^7$ to $3.9x10^7$ FLOPs (using terminology from (He et al., 2015)) for a ResNet-50. The network itself has $3.8x10^9$ FLOPs in total. The range in our count comes from how we measure *Log* and *Error Function* operations that are computationally expensive compared to more standard ops and whose implementations vary. We estimate the worst case from available software instantiations. Most of the work comes from the initial computation of statistics over activation tensors. This ranges from $9.2x10^6$ to $3.7 x 10^7$ FLOPs. In total, this gives us an overhead of 0.3% to 1.0% relative to a ResNet-50 forward pass. All gradient methods have a nominal overhead of at least 100%. A breakdown of the FLOPs per layer and component can be seen in Table 6 in the *appendix*.

Compared to any method which requires a full backward pass, such as gradient methods, our solution is nominally between 97*x* and 344*x* faster for non-SmoothGrad techniques, which according to (Hooker et al., 2018) performs poorly on ROAR/KAR scores. We are between 1456*x* and 5181*x* faster than a 15-iteration SmoothGrad implementation that yields the competitive results in Table 2. 15 iterations as well as other parameters was chosen by (Hooker et al., 2018) who describe this selection in more detail.

The memory footprint of our method is minuscule. Computation over tensors can be done inline which leaves the largest storage demand being the retention of 2D saliency maps. This is increased slightly by needing to store one extra $112x112$ image during bilinear up-sampling. Peak memory overhead related to data is about 117 *kilobytes* per $224x224$ input image.

## 4.2 Usage with a Class Activation Map

Our method does not have a class activation map (Selvaraju et al., 2017; Chattopadhyay et al., 2018; Omeiza et al., 2019) in the current implementation. This is because what we have is more akin to Guided Backprop which Grad-CAM etc. combine with their class activation map to improve their pixel level accuracy. The overhead for the class activation map itself is not large since in practice, it involves computing gradients over the last few network layers only. This makes Guided Backprop

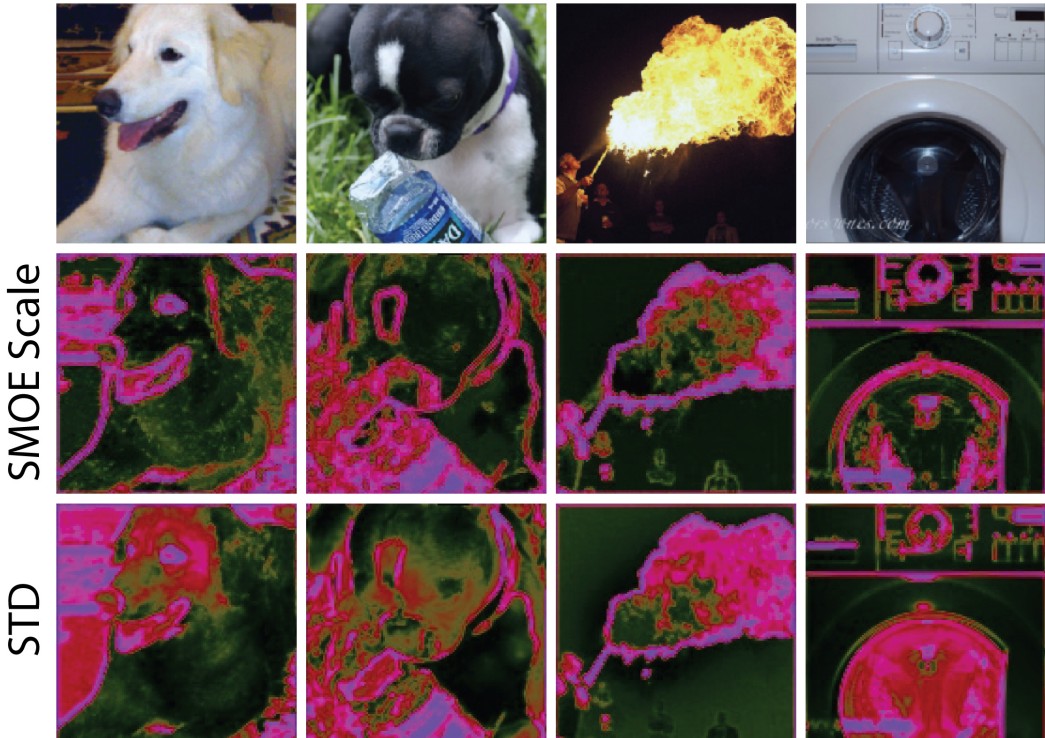

Figure 6: These are examples of the first level saliency maps from SMOE Scale and Standard Deviation. It is common for both std and truncated normal entropy to flood in areas with modest texture. This may explain why difference scores for these two methods are at or below the performance of a random saliency map.

.

the most expensive computational part. By replacing it with our method, accuracy should increase as per Table 2 while dramatically reducing both computational and memory overhead.

### 4.3 SMOE Scale is The Most Robust Measure

SMOE Scale is the only metric without a failure case among all the statistics we tested. It is the only static that reliably scores in the top half of *all* the ones we tested. For Image Net and Places, it is the only one always in the top three. All statistics except for SMOE Scale and Shannon Entropy have at least one layer where they have a difference score at or below zero. This means they are as accurate as a random saliency map for at least one condition. SMOE Scale is the most *robust* statistic to use in terms of expected accuracy. The next highest scoring statistics, standard deviation and truncated normal entropy, are no better than a random on layer 1. Figure 6 shows why this may be. It is important to note that layer 1 contains the finest pixel details and should be expected to be important in rendering visual detail in the final combined saliency map.

## 5 Conclusion

We have created a method of XAI saliency which is extremely efficent and is quantitatively comparable or better than several popular methods. It can also be used to create a saliency map with interpretability of individual scale layers. Future work includes creating a class specific activation map and expanding it to non-DNN architectures. We are currently testing our class activation map integration *Fast-CAM* and will introduce it in a future work.

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

# A APPENDIX

## A.1 DERIVATION OF SMOE SCALE

The maximum likelihood estimator of scale in the Gamma probability distribution is given as:

$$\hat{\theta} = \frac{1}{kn} \sum_{i=1}^{n} x_i \tag{10}$$

This requires the additional iterative estimation of the shape parameter $k$ starting with an estimate $s$:

$$s = \ln\left(\frac{1}{n} \sum_{i=1}^{n} x_i\right) - \frac{1}{n} \sum_{i=1}^{n} \ln(x_i) \tag{11}$$

Then we get to within 1.5% of the correct answer via:

$$k \approx \frac{3 - s + \sqrt{(s-3)^2 + 24s}}{12s} \tag{12}$$

Then we use the Newton-Ralphson update to finish:

$$k \leftarrow k - \frac{\ln(k) - \psi(k) - s}{\frac{1}{k} - \psi'(k)} \tag{13}$$

But we can see application of Eqs 12 and 13 is monotonic. This is also apparent from the example which we can see in Figure 7.

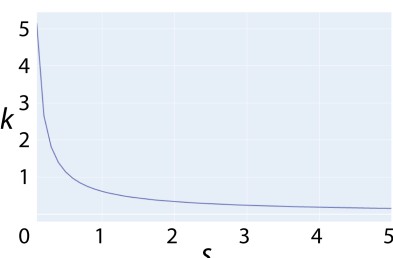

Figure 7: A plot of the resulting $k$ values from input $s$ values in the gamma probability distribution maximum likelihood estimation. It is monotonic and reciprocal.

$k$ is SMOE to $\frac{1}{s}$, so we rewrite Eq 10 with the reciprocal of $k$ and optionally use the more efficient $log_2$ as:

$$\hat{\theta}_{SMOE} = \frac{1}{n} \cdot \sum_{i=1}^{n} x_i \cdot \left[log_2\left(\frac{1}{n} \cdot \sum_{i=1}^{n} x_i\right) - \frac{1}{n} \cdot \sum_{i=1}^{n} log_2(x_i)\right] \tag{14}$$

This then simplifies to:

$$\hat{\theta}_{SMOE} = \frac{1}{n} \cdot \sum_{i=1}^{n} \mu \cdot \log_2\left(\frac{\mu}{x_n}\right) \tag{15}$$

We can see the results this gives with different kinds of data in Table 3

Table 3: **Examples of SMOE Scale results given different data**. This shows in particular when log variance and standard deviation give similar or diverging results. It is easier to see how SMOE Scale as a measure or variance is proportional to the mean. So, if we have lots of large values in an output, we also need them to exhibit more variance relative to the mean activation.

| Input Values | | | | | | | | | Mean | STD | SMOE Scale |
|---|---|---|---|---|---|---|---|---|---|---|---|
| 0.5 | 1 | ... | 0.5 | 1 | 0.5 | 1 | 0.5 | 1 | 0.75 | 0.25 | 0.064 |
| 1 | 2 | ... | 1 | 2 | 1 | 2 | 1 | 2 | 1.5 | 0.5 | 0.127 |
| 2 | 4 | ... | 2 | 4 | 2 | 4 | 2 | 4 | 3 | 1 | 0.255 |
| 1 | 2 | ... | 1 | 2 | 1 | 2 | 1 | 2 | 1.5 | 0.5 | 0.127 |
| 2 | 3 | ... | 2 | 3 | 2 | 3 | 2 | 3 | 2.5 | 0.5 | 0.074 |
| 2 | 4 | ... | 2 | 4 | 2 | 4 | 2 | 4 | 3 | 1 | 0.255 |
| 0.6125 | 1.8375 | ... | 0.6125 | 1.8375 | 0.6125 | 1.8375 | 0.6125 | 1.8375 | 1.225 | 0.6125 | 0.254 |

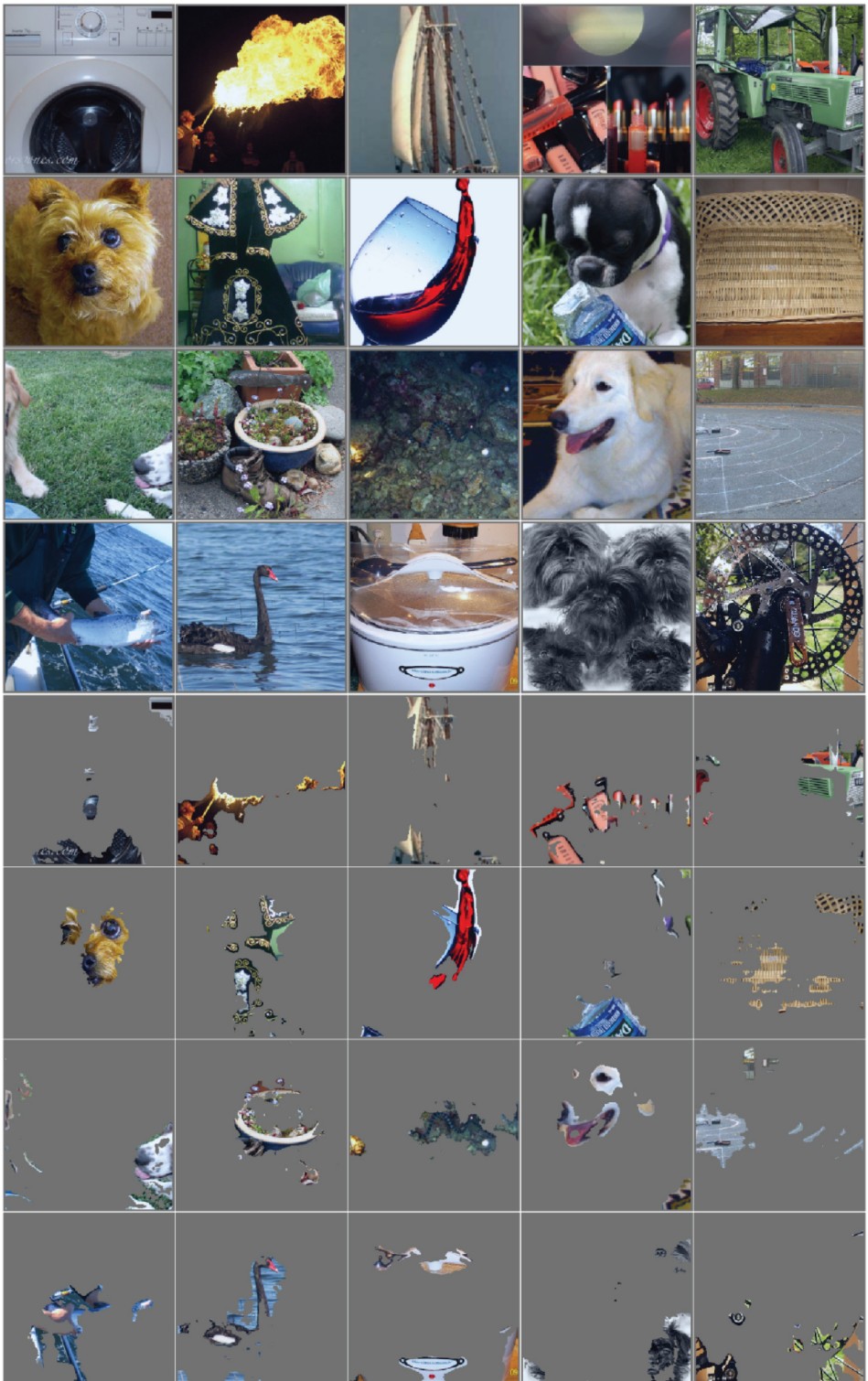

Figure 8: These are the last mini-batch images in our *GPU:0* buffer when running the ImageNet validation set. The top images are the original input images and the ones on the bottom are 10% KAR images of the most salient pixels. These are images used when computing KAR scores.

A.3    MORE EXAMPLES OF COMBINED SALIENCY MAPS

Figure 9: These are more examples of combined saliency maps using the same images that appear in Figure 8. These images are *not* alpha blending with the original. Above each image is the ground truth label, while the label the network gave it is below. This was auto-generated by our training scripts.

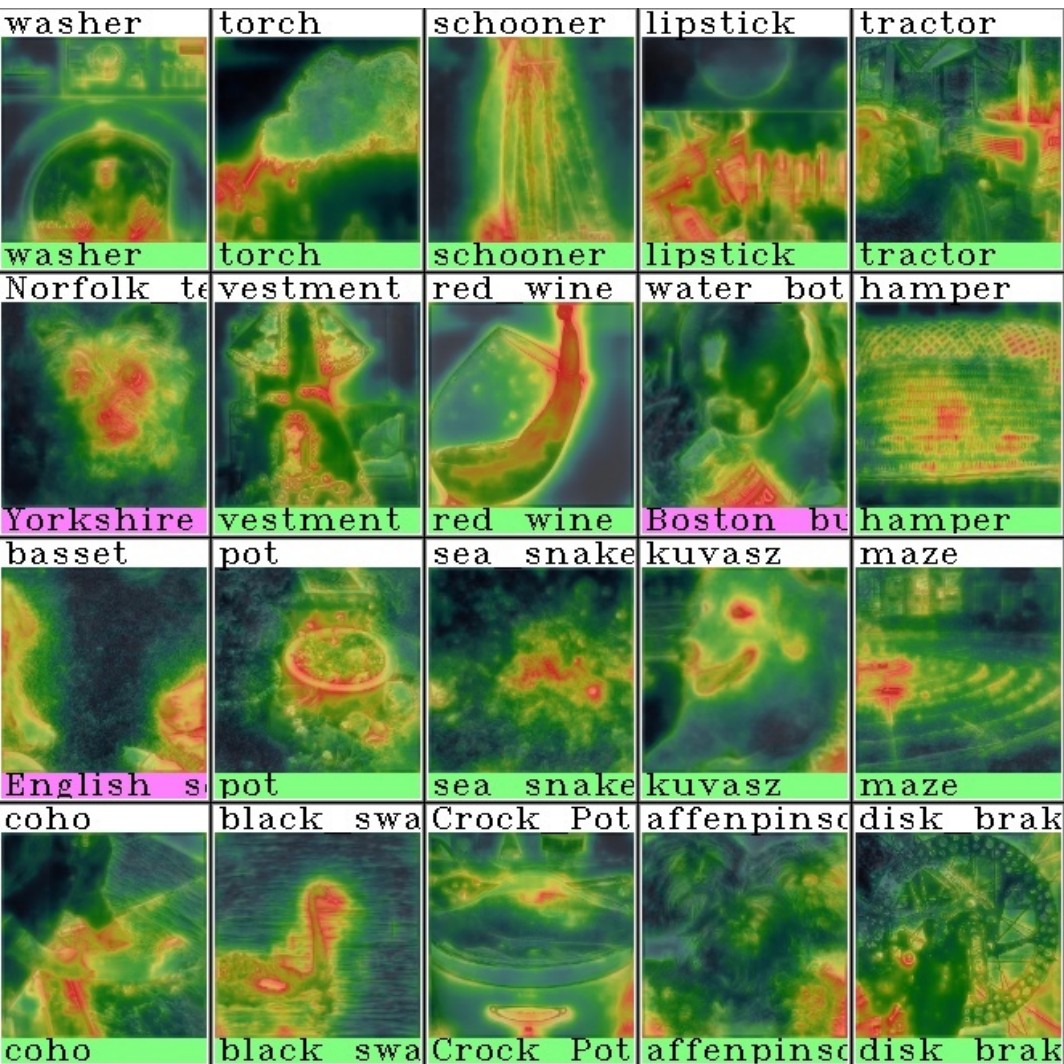

Figure 10: These are the same as Figure 9 except with the original image gray scaled and alpha blended at 25%.

## A.4 MORE EXAMPLES OF LOVI SALIENCY MAPS

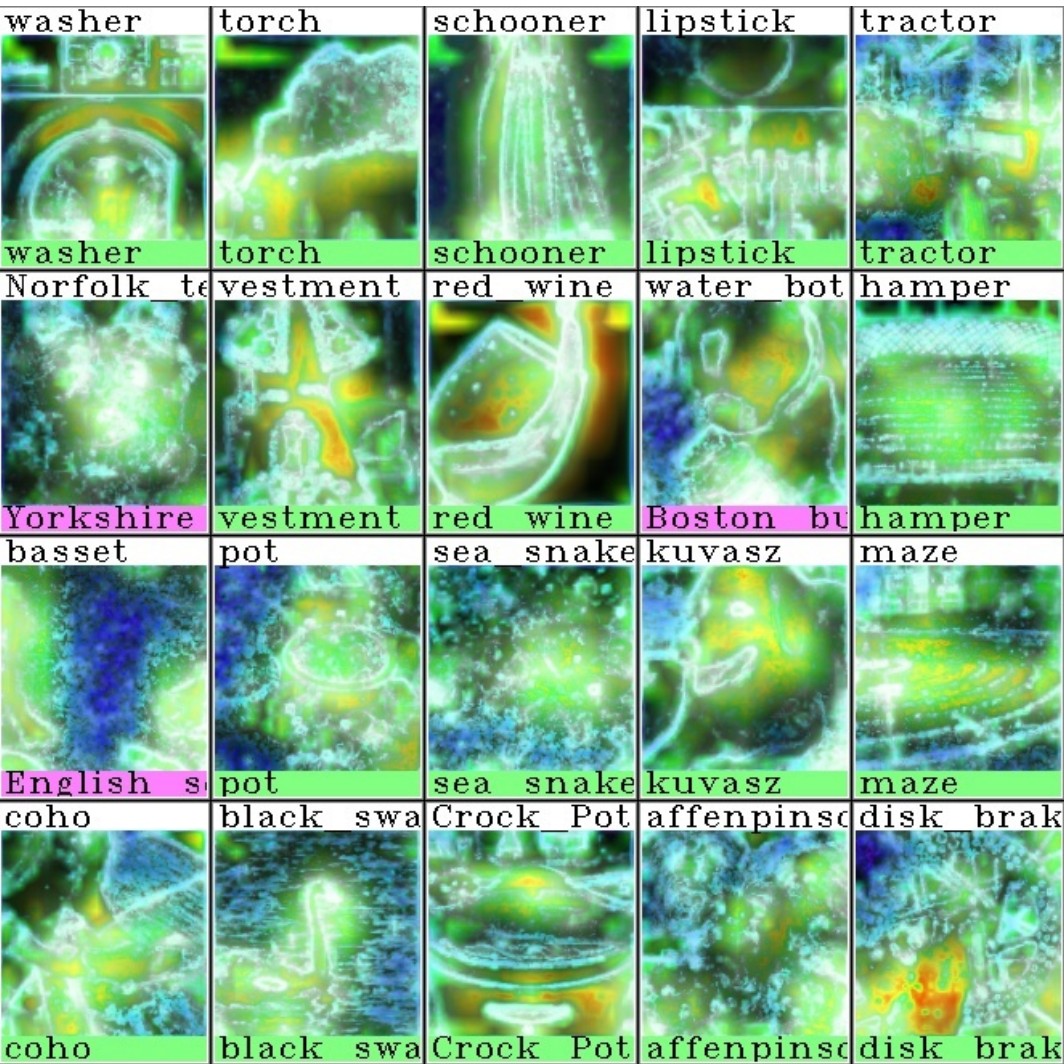

Figure 11: These are more examples of visualizing multiple saliency maps using the same images that are in Figure 8. These images are *not* alpha blending with the original.

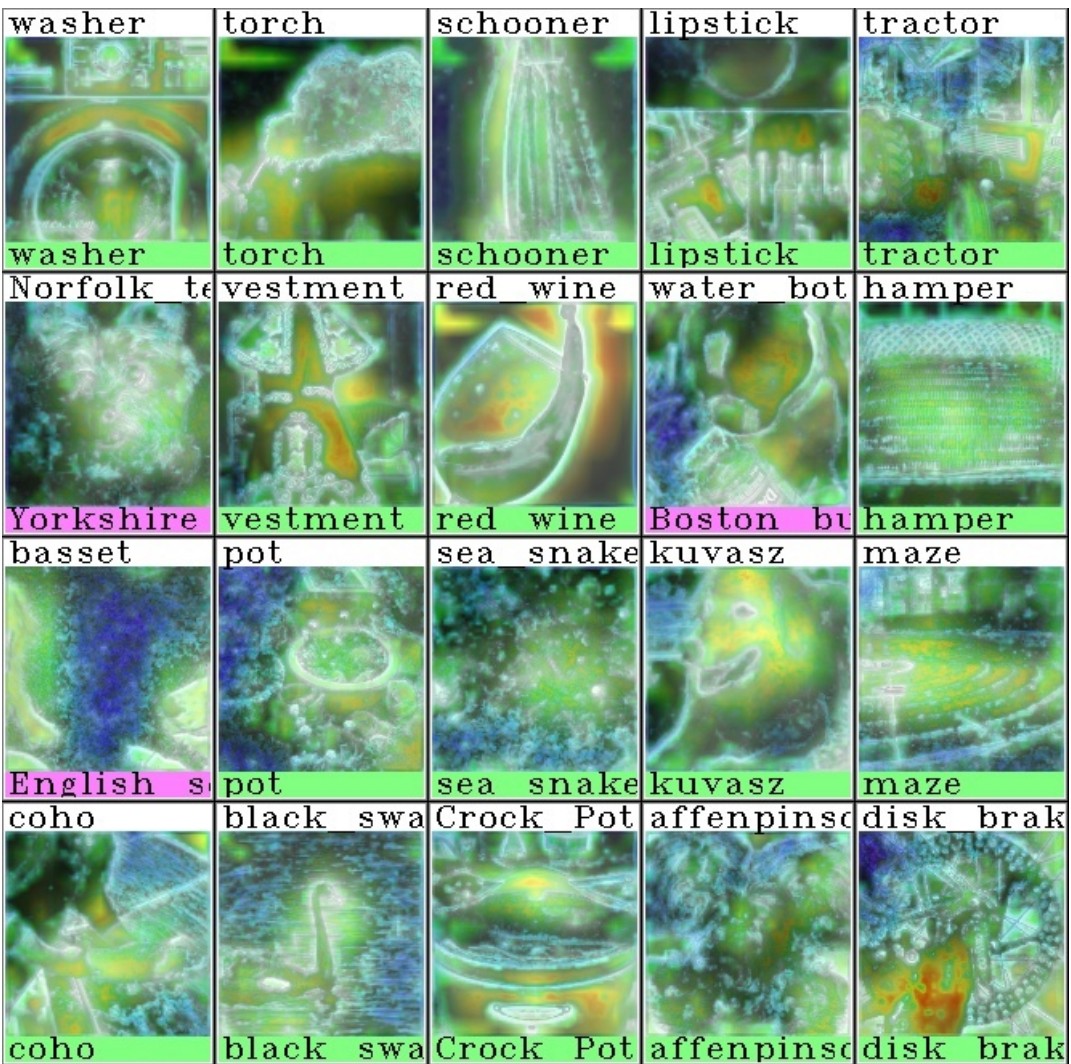

Figure 12: These are the same as Figure 11 with the original image gray scaled and alpha blended at 25%.

## A.5 COMPARING DIFFERENT EFFICIENT STATISTICS IN MORE DETAIL

This subsection shows the raw scores for each statistic over each dataset.

Table 4: **KAR and ROAR results per dataset**. This is a more detailed version of Table 1 and Figure 4. The differing effects of the distribution of data in the three sets seems to manifest itself in the L5 scores whereby the more concentrated the information is spatially, the better the ROAR/L5 score seems to be.

| Method | KAR Kept Percent | | | | | ROAR Removed Percent | | | | |
|---|---|---|---|---|---|---|---|---|---|---|
| | L1 | L2 | L3 | L4 | L5 | L1 | L2 | L3 | L4 | L5 |
| **ImageNet** | | | | | | | | | | |
| Random | 66.42 | 61.28 | 50.67 | 40.81 | 42.98 | 73.48 | 72.41 | 68.90 | 64.63 | 66.04 |
| SMOE Scale | 56.61 | 50.69 | 51.25 | 46.40 | 63.00 | 44.48 | 44.81 | 36.35 | 33.88 | 21.15 |
| STD | 51.84 | 50.73 | 50.72 | 46.16 | 62.82 | 45.78 | 42.74 | 36.17 | 34.41 | 22.88 |
| Mean | 53.21 | 40.34 | 50.88 | 46.85 | 62.56 | 52.66 | 64.10 | 37.85 | 34.15 | 19.19 |
| Shannon Ent | 55.43 | 45.69 | 50.89 | 47.17 | 61.18 | 44.74 | 51.38 | 38.73 | 35.78 | 18.07 |
| Log Normal Mean | 55.98 | 32.28 | 51.08 | 47.21 | 62.02 | 57.20 | 68.22 | 44.42 | 34.98 | 18.50 |
| Log Normal ENT | 53.01 | 42.52 | 51.13 | 46.85 | 62.26 | 47.92 | 62.64 | 38.73 | 34.50 | 18.91 |
| Trunc Normal Mean | 50.67 | 49.69 | 50.69 | 43.52 | 62.87 | 46.88 | 49.76 | 35.44 | 37.58 | 20.92 |
| Trunc Normal Std | 50.66 | 51.02 | 50.60 | 42.54 | 62.97 | 46.78 | 43.70 | 35.68 | 38.18 | 21.57 |
| Trunc Normal Ent | 50.84 | 50.62 | 50.57 | 43.63 | 62.97 | 46.92 | 45.48 | 35.56 | 37.64 | 21.25 |
| **Best** | 56.61 | 51.02 | 51.25 | 47.21 | 63.00 | 44.48 | 42.74 | 35.44 | 33.88 | 18.07 |
| **Worst** | 50.66 | 32.28 | 50.57 | 42.54 | 61.18 | 57.20 | 68.22 | 44.42 | 38.18 | 22.88 |
| **Places** | | | | | | | | | | |
| Random | 57.20 | 53.59 | 47.83 | 41.26 | 39.45 | 60.77 | 60.25 | 58.14 | 56.41 | 55.26 |
| SMOE Scale | 49.76 | 45.67 | 46.39 | 40.57 | 53.50 | 44.35 | 44.61 | 39.80 | 40.26 | 27.94 |
| STD | 47.15 | 44.75 | 46.28 | 39.41 | 53.53 | 46.41 | 43.69 | 39.28 | 41.38 | 29.12 |
| Mean | 47.93 | 40.38 | 45.94 | 41.10 | 52.33 | 50.66 | 56.58 | 41.26 | 39.90 | 27.38 |
| Shannon Ent | 48.80 | 43.20 | 45.92 | 41.31 | 50.62 | 41.97 | 49.28 | 42.93 | 39.98 | 27.06 |
| Log Normal Mean | 50.05 | 35.87 | 46.23 | 41.45 | 51.67 | 52.21 | 57.91 | 45.17 | 39.73 | 26.88 |
| Log Normal ENT | 47.77 | 41.41 | 46.02 | 41.39 | 52.25 | 48.96 | 56.34 | 41.91 | 39.68 | 27.00 |
| Trunc Normal Mean | 46.25 | 44.76 | 46.12 | 38.08 | 53.18 | 46.92 | 46.92 | 48.83 | 42.09 | 28.06 |
| Trunc Normal Std | 45.96 | 45.35 | 46.38 | 37.61 | 53.38 | 46.41 | 46.76 | 44.86 | 42.43 | 28.68 |
| Trunc Normal Ent | 46.06 | 45.01 | 46.38 | 37.57 | 53.15 | 46.67 | 46.67 | 38.85 | 42.09 | 28.11 |
| **Best** | 50.05 | 45.67 | 46.39 | 41.45 | 53.53 | 41.97 | 43.69 | 38.85 | 39.68 | 26.88 |
| **Worst** | 45.96 | 35.87 | 45.92 | 37.57 | 50.62 | 52.21 | 57.91 | 48.83 | 42.43 | 29.12 |
| **COWC** | | | | | | | | | | |
| Random | 65.05 | 57.43 | 52.30 | 64.31 | 65.55 | 77.38 | 75.44 | 71.11 | 78.25 | 77.32 |
| SMOE Scale | 64.19 | 63.02 | 71.05 | 62.87 | 80.65 | 45.16 | 44.09 | 43.97 | 62.78 | 59.49 |
| STD | 60.36 | 61.95 | 70.89 | 64.57 | 80.55 | 44.12 | 43.52 | 44.10 | 60.69 | 59.73 |
| Mean | 63.82 | 59.90 | 73.20 | 61.83 | 80.54 | 45.79 | 59.24 | 44.61 | 64.02 | 58.86 |
| Shannon Ent | 62.64 | 63.78 | 73.29 | 60.99 | 78.77 | 46.50 | 44.23 | 46.42 | 68.31 | 57.98 |
| Log Normal Mean | 66.37 | 46.05 | 72.89 | 60.21 | 80.38 | 48.98 | 71.02 | 46.69 | 67.19 | 58.16 |
| Log Normal ENT | 63.23 | 62.78 | 73.26 | 60.99 | 80.44 | 45.10 | 57.12 | 44.92 | 65.61 | 58.56 |
| Trunc Normal Mean | 60.00 | 63.35 | 72.08 | 65.62 | 80.54 | 42.98 | 44.44 | 44.15 | 61.60 | 59.45 |
| Trunc Normal Std | 59.52 | 63.74 | 71.58 | 65.58 | 80.54 | 42.76 | 43.45 | 43.98 | 62.07 | 59.81 |
| Trunc Normal Ent | 59.79 | 63.48 | 71.77 | 65.68 | 80.59 | 43.04 | 43.80 | 43.89 | 61.79 | 59.89 |
| **Best** | 66.37 | 63.78 | 73.29 | 65.68 | 80.65 | 42.76 | 43.45 | 43.89 | 60.69 | 57.98 |
| **Worst** | 59.52 | 46.05 | 70.89 | 60.21 | 78.77 | 48.98 | 71.02 | 46.69 | 68.31 | 59.89 |

A.6    COMBINED KAR AND ROAR SCORES WITH MORE DETAIL

This subsection shows the raw scores for each ROAR and KAR mask. We also added the non-SmoothGrad methods so we can see how much of an improvement it makes.

Table 5: **Combined KAR and ROAR scores for several methods**. This is a more detailed version of Table 2 and Figure 5. The top six rows show several popular methods with and without *Squared SmoothGrad* applied to give optimal results. These are taken from (Hooker et al., 2018). *Prior Layer Weights* means we applied layer weights based on the prior determined accuracy of the layer saliency map. We include our top three scoring methods. The SMOE Scale method outperforms the three baseline methods on KAR. It outperforms Guided Backprop and Integrated Gradients on ROAR as well as overall. The Gradient method is best overall, but as we discussed, it is much more expensive to compute.

| | KAR Kept Percent | | | | | ROAR Removed Percent | | | | |
|---|---|---|---|---|---|---|---|---|---|---|
| Method | 10% | 30% | 50% | 70% | 90% | 10% | 30% | 50% | 70% | 90% |
| Rand | 63.53 | 67.06 | 69.13 | 71.02 | 72.65 | 72.65 | 71.02 | 69.13 | 67.06 | 63.53 |
| Gradient | 67.63 | 71.45 | 72.02 | 72.85 | 73.46 | 72.94 | 72.22 | 70.97 | 70.72 | 66.75 |
| Guided Backprop | 71.03 | 72.45 | 72.28 | 72.69 | 71.56 | 72.29 | 71.91 | 71.18 | 71.48 | 70.38 |
| Integrated Grad. | 70.38 | 72.51 | 72.66 | 72.88 | 73.32 | 73.17 | 72.72 | 72.03 | 71.68 | 68.20 |
| Gradient -w- SmoothGrad Sq. | 63.25 | 69.79 | 72.20 | 73.18 | 73.96 | 69.35 | 60.28 | 41.55 | 29.45 | 11.09 |
| Guided Backprop -w- SmoothGrad Sq. | 62.42 | 68.96 | 71.17 | 72.72 | 73.77 | 69.74 | 60.56 | 52.21 | 34.98 | 15.53 |
| Integrated Grad. -w- SmoothGrad Sq. | 67.55 | 68.96 | 72.24 | 73.09 | 73.80 | 70.76 | 65.71 | 58.34 | 43.71 | 29.41 |
| SMOE Scale + Prior Layer Weights | 65.44 | 72.14 | 74.28 | 74.51 | 75.01 | 70.40 | 60.33 | 48.48 | 34.23 | 17.72 |
| SMOE Scale + Layer Weights [1,...,5] | 65.76 | 72.60 | 73.97 | 74.53 | 74.94 | 70.28 | 60.93 | 48.73 | 35.66 | 18.01 |
| SMOE Scale + Layer Weights [1,...,1] | 66.13 | 72.28 | 73.72 | 74.52 | 74.97 | 71.28 | 63.58 | 52.85 | 38.74 | 19.72 |
| Normal Std + Prior L. Weights | 65.48 | 72.17 | 73.93 | 74.62 | 74.67 | 69.98 | 60.39 | 48.75 | 34.63 | 18.13 |
| Trunc Normal Ent + Prior L. Weights | 65.45 | 72.38 | 74.10 | 74.40 | 74.75 | 69.85 | 60.08 | 48.05 | 34.32 | 18.37 |

A.7   OPERATIONS COMPUTATION

Table 6: **FLOPs for each layer**. This is the breakdown of FLOPs for each layer. *Log* and *Error Function* are counted as one operation in this example. **SMOE Ops** is how many operations it takes to compute the initial saliency map using the SMOE Scale statistic. **Norm Ops** is the number of operations needed to normalize the saliency map. **Combine Ops** is the number of ops needed to upsample and combine each saliency map.

| | Layer Dimensions | | | FLOPS | | |
| Layer | Channels | Size H | Size W | SMOE Ops | Norm Ops | Combine Ops |
|---|---|---|---|---|---|---|
| Layer 1 | 64 | 112 | 112 | 3223808 | 150528 | 225792 |
| Layer 2 | 256 | 56 | 56 | 3214400 | 37632 | 338688 |
| Layer 3 | 512 | 28 | 28 | 1606416 | 9408 | 338688 |
| Layer 4 | 1024 | 14 | 14 | 803012 | 2352 | 338688 |
| Layer 5 | 2048 | 7 | 7 | 401457 | 588 | 338688 |
| Total | | | | 9249093 | 200508 | 1580544 |

