# OpenReview forum: "Efficient Saliency Maps for Explainable AI"
_ICLR.cc/2020/Conference — Reject_

### Official Review · AnonReviewer3 · 2019-10-18
**Official Blind Review #3**

**Rating:** 6

**Review:**

This paper presents a method for creating saliency maps reflecting what a network deems important while also proposing an interesting method for visualizing this. The central premise for the method of characterizing relative importance of information represented by the network is based on an information theoretic measure. A variety of results are presented to examine the impact of keeping or removing information deemed important by this measure and a comparison is made to existing approaches as a justification for the proposed methods.
I find that the proposed method appears to be theoretically sound and is interesting in revealing differences to other information theoretic methods especially in the early layers. The relative similarity in later layers is also an interesting observation and one that is relatively hard to characterize.
Perhaps the strongest case for the proposed method comes from the keep accuracy subject to SMOE scale. The removal accuracy is a bit less convincing but appears to be a fair and honest evaluation. Moreover, the LOVI scheme for visualization is interesting in itself and I found this aspect of the work to be thought provoking with respect to how such methods are examined from a qualitative perspective.

**Experience Assessment:**

I have read many papers in this area.

**Review Assessment: Checking Correctness Of Derivations And Theory:**

I assessed the sensibility of the derivations and theory.

**Review Assessment: Checking Correctness Of Experiments:**

I assessed the sensibility of the experiments.

**Review Assessment: Thoroughness In Paper Reading:**

I read the paper at least twice and used my best judgement in assessing the paper.

---

### Official Review · AnonReviewer2 · 2019-10-22
**Official Blind Review #2**

**Rating:** 3

**Review:**

The paper presents a new approach, SMOE scale, to extract saliency maps from a neural network. The approach is deemed as efficient, because it does not require one or multiple backward passes, as opposed to other approaches based on gradients. The main idea is to process the output activation tensors of a few selected layers in a deep network using an operator combining mean and std.dev. called SMOE scale. The result of this operator can be combined through different scales to obtain a global saliency map, or visualized in such a way that shows the consistency of saliency maps at different scales. Experiments show some improvement against traditional gradient-based approaches.

This paper is interesting in that it provides a different way to extract saliency maps from a network. The visualization at multiple scales is also nice and while I do not perfectly agree with the HSV encoding in Fig.2, I do see the potential. Being efficient is also a nice to have. There are three issues, however, which limits the novelty of the paper. First, the SMOE metric does not seem to bring much improvement compared to simple metrics. Second, the few comparisons made against other methods do not reveal a significant improvement. Third, at core, the paper suggests that the "high efficiency" of this approach is one of its main advantages, a statement I do not forcibly agree with. More details follow.

For the first element, we have to consider the paper as the combination of two things. 1) the use of activation maps as source of salient information, and 2) the way we should process these activation maps. 1) is relatively straightforward, so the core of the contribution should lie in 2). However, while the SMOE scale method definition (eq.2) is sound, it does not bring valuable improvement compared to other "trivial" metrics, like standard deviation. For instance, Fig.4 caption tells that "SMOE Scale differentiates itself the most early on in the network", but it is actually only for the very first scale layer. At every other scale, standard deviation (for instance) is at least as good. Same thing can be said about Table 4 in appendix, and also about Table 2 (and the scores of Trunc. Normal Entropy). Overall, while SMOE is indeed novel, it is not highly convincing.

On a side note about the SMOE description, I did not find the list of "conditions and assumptions" at the beginning of Sec. 2.1. It looks more like an after-thought over which the proposed method coincidentally fits. Moreover, point 3 is kind of conflicting in its formulation.

For the second element, the improvements in KAR and ROAR scores are quite minimal. It does seem to have an edge on KAR score, but not by a huge amount. Additionally, the methods compared are relatively old. To give just two examples of missing new techniques, Smooth Grad-CAM++ (Omeiza et al.) or even Grad-CAM++ (Chattopadhay et al.) would presumably obtain better performances. Moreover, Smooth Grad-CAM++ allows to target a particular feature map or even a specific neuron, which makes it even more relevant to this work.

Finally, a note about efficiency. Generally speaking, I agree that it is always good to be more efficient. However, I fail to see the high importance given to efficiency for this particular problem. Sure, gradient-based approaches are probably not suitable to online, in-network applications, but is it an important requisite? Computing saliency maps on a subset of the dataset once a few epochs already gives a good idea of what the network is doing. In any case, since these saliancy maps are intended for human use, I am not convinced about the importance of computing them for each training example at each epoch. Overall, in my opinion, being efficient at generating saliency maps is a nice to have, but not much more.

Some general comments:
- In Sec. 2.2, the last "con" seems a bit out of place. This could be applied to pretty much anything.
- Sec. 2.3 is interesting, but the explanations are convoluted. In essence, what should be said is 1) value encompasses the importance of a pixel, 2) saturation presents the max-min of the distribution and 3) hue shows the position in the network. Also, superimposing these HSV maps over gray scale version of the image like is Fig.2 is difficult to analyze because the "gray" of the image can be confused with the saturation channel.
- On p.2, "this is proceeded" -> "this is preceded"?

In summary, this paper presents a nice way of generating saliency maps from activations inside a network. However, the comparison to other approaches does not show a clear advantage, and the related work (and experiments) lacks recent techniques. I would thus ask this general question: what is the "selling point" of this method? The current focus on efficiency does not convince me. That being said, there are no clear flaws in the paper, so I am open to reconsider my assessment if improvements are made to the paper (better descriptions, comparison with more recent techniques, experimental justification for SMOE instead of simpler approaches, etc.).

On a final note, there is also the Score-CAM approach (Wang et al.) that looks similar (in the idea of using activation maps). I did not consider it in this review since it was published a few weeks ago on Arxiv, but it could be interesting to discuss it nevertheless.

**Experience Assessment:**

I have read many papers in this area.

**Review Assessment: Checking Correctness Of Derivations And Theory:**

I carefully checked the derivations and theory.

**Review Assessment: Checking Correctness Of Experiments:**

I carefully checked the experiments.

**Review Assessment: Thoroughness In Paper Reading:**

I read the paper thoroughly.

---

> ### Author Response · Authors · 2019-11-09
> **Thank You**
>
> We would like to thank the reviewer for their helpful suggestions and comments. We are posting our responses first and will add the revised paper soon. Traditionally, these come at the same time. However, given the spirit of this open review process, it seems prudent to allow time for discussion. Hopefully we are not misinterpreting the intent of the conference organizers.
>
> Much of our response as well as edits to the existing manuscript involve questions regarding efficiency. Indeed, it’s in the title. Given the comments, it is obvious that we did not give it the proper treatment. Our paper will add usage examples to the introduction and a breakdown of both time and memory efficiency in the discussion.
>
> More comments will follow ;)

---

> ### Author Response · Authors · 2019-11-11
> **Response 1**
>
>
> ##########################################################################
> Second, the few comparisons made against other methods do not reveal a significant improvement.
> ##########################################################################
>
> (A)	The proposed method, if used to replace Guided Backprop, will increase the speed of both Grad-CAM and Grad-CAM++ by two orders of magnitude, lower their memory footprint significantly and probably noticeably increase accuracy.  We invite the reviewer to please consider our more detailed efficiency calculations and our responses that follow in their evaluation.

---

> > ### Comment · AnonReviewer2 · 2019-11-14
> > **Context**
> >
> > In the context of this sentence, I was talking about performance improvements, as explained in details next. Not that speed and memory footprint do not matter at all of course.

---

> ### Author Response · Authors · 2019-11-11
> **Response 2**
>
>
> ##########################################################################
> Third, at core, the paper suggests that the "high efficiency" of this approach is one of its main advantages, a statement I do not forcibly agree with. More details follow.
> ##########################################################################
>
> We will include a full discussion section to detail time complexity, memory complexity and operation counts. Operations from our method come from three sources. The first is the computation of statistics on tensors in five layers. The second is the normalization of each 2D saliency map. Third we account for the cost of combing the saliency maps.
>
> Ops for our solution ranges from 1.1 x 10^7 to 3.9 x 10^7 FLOPs (using terminology from He 2015 [the original ResNet paper]) for a ResNet-50. The network itself has 3.8 x 10^9 FLOPs in total. The range in our method comes from how one counts Log and Error Function operations which are operationally expensive compared to more standard ops.  How they work is apparently a Nvidia trade secret, so we estimate the worst case from available software instantiations. Most of the work comes from the initial computation of statistics over activation tensors. This ranges from 9.2 x 10^6 to 3.7 x 10^7 FLOPs. In total, this gives our model an overhead of 0.3% to 1.0% relative to the network itself. All full backward pass methods have a nominal overhead of 100%. For larger networks, our overhead decreases relatively since it is constant.
>
> Compared to any method which requires a full backward pass, such as gradient methods, our solution is nominally between 97x and 344x faster for non-SmoothGrad techniques, which according to Hooker et al. perform poorly on ROAR/KAR scores. We are between 1456x and 5181x faster than a 15-iteration SmoothGrad implementation that yields the results in figure 5. 15 iterations as well as other parameters was chosen by Hooker et al. who describes this selection in more detail.
>
> The memory footprint of our model is minuscule. Computation over tensors can be done inline which leaves the largest storage demand being the retention of 2D saliency maps. This is increased slightly by needing to store one extra 112x112 image during bilinear up-sampling. Peak memory overhead related to data storage is about 117 kilobytes per 224x224 input image.
>
> We will attach detailed tables to the Appendix showing our work.

---

> ### Author Response · Authors · 2019-11-11
> **Response 3**
>
>
> ##########################################################################
> First, the SMOE metric does not seem to bring much improvement compared to simple metrics.
>
> For the first element, we have to consider the paper as the combination of two things. 1) the use of activation maps as source of salient information, and 2) the way we should process these activation maps. 1) is relatively straightforward, so the core of the contribution should lie in 2). However, while the SMOE scale method definition (eq.2) is sound, it does not bring valuable improvement compared to other "trivial" metrics, like standard deviation. For instance, Fig.4 caption tells that "SMOE Scale differentiates itself the most early on in the network", but it is actually only for the very first scale layer. At every other scale, standard deviation (for instance) is at least as good. Same thing can be said about Table 4 in appendix, and also about Table 2 (and the scores of Trunc. Normal Entropy). Overall, while SMOE is indeed novel, it is not highly convincing.
> ##########################################################################
>
> (1)	The layer 1 scores appear closer than they are. We will mention that the layer 1 Difference score for the second-best metric, Standard Deviation is 0.0031. This means its performance is almost the same as a random saliency map. SMOE Scale is 0.1051. Only Log Normal Mean and Shannon Entropy score near it. However, those two do not do very well overall.
> (2)	We will add a table to show that a difference of 0.102 between Standard Deviation and SMOE Scale is rather large since for layers 3 through 5, there is never a difference greater than 0.0625 between any of tested methods.
> (3)	We are adding a graphic to show that Standard Deviation has a tendency to flood on the layer 1 maps and fail to catch low level features one would expect to be important. This suggests why Standard Deviation appears to fail on this layer. Truncated normal entropy suffers even more.
> (4)	Layer 1 is important since it carries the finest spatial details for the combined saliency map.
> (5)	SMOE scale is the only metric which does not seem to have a failure case in terms of either scale or dataset.
> (6)	There is a third contribution to consider here. Consider if we did not have the SMOE Scale results, but instead just published the results from the standard deviation. Doesn’t the fact that it does outright better than Guided Backprop and Integrated Gradients (With SmoothGrad) surprise you just a little bit? I know we found that result by itself to be rather unexpected. We thought our efficient method to be worse, but hopefully not too much more so than the three gradient methods. We did not think that a simple weighted average over statistical feature maps should give such results.
>     a.	We may not have elucidated this argument explicitly, but this was part of the reason for our candor with Table 2. We also put it there because honesty is the bedrock of human knowledge. We will make our point about (6) much clearer in the revision.

---

> > ### Comment · AnonReviewer2 · 2019-11-14
> > **Nice addition**
> >
> > Fig. 6 is indeed a nice visualization of the benefits of using SMOE.

---

> ### Author Response · Authors · 2019-11-11
> **Response 4**
>
>
> ##########################################################################
> On a side note about the SMOE description, I did not find the list of "conditions and assumptions" at the beginning of Sec. 2.1. It looks more like an after-thought over which the proposed method coincidentally fits. Moreover, point 3 is kind of conflicting in its formulation.
> ##########################################################################
>
> These conditions are indeed what we were after. If the review was not anonymous, it would be easy to show that these are things we are interested in 😉
>
> Condition 3 reflects a mid-project addendum based on observations and further discussions. It got edited too many times by the different authors and author 1 was worried it still didn’t sound right. It might make more sense to be split into two statements. At any rate, we will reword it.

---

> > ### Comment · AnonReviewer2 · 2019-11-14
> > **The issue with "conditions list"**
> >
> > I would certainly believe that these conditions were the one which drove the authors in the conception of this work. My issue was more about their usefulness in the context of the paper.
> >
> > 1) these are not objective nor motivated conditions (e.g. something like "it must run at 30 FPS on the hardware X because we use it real time")
> > 2) they cannot be reused by another work (or, conversely, they could be reused by any other work, since they broadly apply to pretty much everything)
> >
> > So overall, what does bring this enumeration to the paper? My personal feeling is not much, and that's what I tried to express in my initial review.

---

> ### Author Response · Authors · 2019-11-11
> **Response 5**
>
>
> ##########################################################################
> For the second element, the improvements in KAR and ROAR scores are quite minimal. It does seem to have an edge on KAR score, but not by a huge amount.
> ##########################################################################
>
> This needs to be taken in light of the rather large efficiency gain it obtains. Our reply contains a list of situations where efficiency is very important. Our results show that one can confidently use our solution and get at least slightly better results than other widely used methods which are unfeasible to use.

---

> ### Author Response · Authors · 2019-11-11
> **Response 6**
>
>
> ##########################################################################
> Additionally, the methods compared are relatively old. To give just two examples of missing new techniques, Smooth Grad-CAM++ (Omeiza et al.) or even Grad-CAM++ (Chattopadhay et al.) would presumably obtain better performances. Moreover, Smooth Grad-CAM++ allows to target a particular feature map or even a specific neuron, which makes it even more relevant to this work.
> ##########################################################################
>
>
> (A)	These are excellent works. Smooth Grad-CAM++ is a work that is new to us. It looks like it was published to ArXiv just two months before our paper submission. Thank you for pointing it out to us.
> (B)	If we Look at the source for Grad-CAM++, we can see it uses Guided Backprop in its core, the same as Grad-CAM. From Grad_CAM_plus_plus, “line 25: # Guided backpropagtion back to input layer”. Given this, then figure 5 applies to it the same as for Grad-CAM. We mentioned in the discussion the distinction between the Class Activation Map and the base saliency map provided by such methods as Guided Backprop. We also mentioned that our technique might be able to serve as a basis for another x-CAM method. What we present in our work, would make Grad-CAM++ much faster and more accurate.
> (C)	The ROAR/KAR method from Hooker et al. we use to evaluate our technique was posted Mar 2019. A very recent work was indeed critical for our results.
>     a.	This work unfortunately got buried in a workshop, we highly recommend taking a look at it. This is the best way we have seen to quantitatively compare explainable AI saliency methods. Fortunately, the author has been actively proselytizing the work, which is how we became aware of it.
> (D)	Two methods we compare to, SmoothGrad and Integrated Gradients are only two years old. Machine learning research moves fast these days, but I would still consider these to be recent.

---

> ### Author Response · Authors · 2019-11-11
> **Response 7**
>
>
> ##########################################################################
> Finally, a note about efficiency. Generally speaking, I agree that it is always good to be more efficient. However, I fail to see the high importance given to efficiency for this particular problem. Sure, gradient-based approaches are probably not suitable to online, in-network applications, but is it an important requisite?
> ##########################################################################
>
> We are adding examples and details. The saliency methods where we have seen quantitative results of efficacy have at least one of three components which create efficiency related issues we discuss below. These components are:
>
> (1)	Usage of gradient data.
> (2)	Usage of multiple passes through the network.
> (3)	Usage of saliency network training.
>
> Below are examples where our method is efficient enough to work well, but other methods are not. The discussion will show a detailed breakdown of our method’s efficiency (see further below). These are divided by efficiency type. The introduction will contain this information, but organized in a standard way, not just a long list.
>
> 1.	Examples of efficiency related to time complexity: This affects methods which use gradient data and multiple network passes. Some saliency network training models may be affected, but we know of some which are not. Examples are:
>     a. If a person is using real-time vision applications where embedded hardware is sometimes barely fast enough. Doing an extra backward pass in the field can lead to a dropped frame each time it is computed.
>         i. Frame drop can also affect data recording.
>     b. In robotics, engineers typical like to see real-time feedback while the robot is running. Per (1.a.), methods that require full backward gradients are less feasible due to irritating frame drop.
>         i. Answering questions like “Why did the robot run into the wall?”, “What was it looking at?”, or “Can I quickly tweak a parameter and get it to behave?” are interactive and iterative in practice, so real-time diagnostics and feedback provided by efficient saliency maps are crucial.
>     c. Time efficiency is important in cloud computing where GPU usage translates into dollars (or carbon).
>     d. Another example is DNNs that use saliency as part of their pipeline. Hypothetically, this means much faster training.
> 2.	Examples of efficiency related to memory complexity: This affects methods which use gradients and may affect any method using a trained saliency network. This is a much more difficult issue to overcome and gradient methods in particular would be untenable in these scenarios since the information required for a backward pass is rather large. These are examples using deployed, pre-trained networks.
>     a. Processing of very large satellite images may preclude the storage of information required to compute gradients.
>     b. Deployed industrial inspection systems frequently have lower cost GPUs with less memory. These are sufficient for running a model but not training one. When these systems fail in production, engineers want to see what is going on. It’s nice if diagnostic information is presented online.
>     c.	Many embedded systems might not only lack sufficient memory in general, but may not even have to ability to store or compute gradient data.
>     d. Methods utilizing a pre-trained saliency network may work in some of these situations so long as they are not too large. However, the addition of even a handful of extra layers may rule out their usage.
> 3.	Issues related to logistical efficiency. This would primarily affect methods which must be pre-trained. It comes from the synchrony of maintaining a trained model in the field and its complementary trained saliency network. It introduces a source of human error. Extra effort must be used to make sure both models are kept in sync.
>
> In summary, we have created the only explainable AI saliency method which will work unconditionally in all these kinds of conditions and has demonstrable and quantitatively good results.

---

> > ### Comment · AnonReviewer2 · 2019-11-14
> > **Efficiency can indeed be useful**
> >
> > I thank the authors for their answer. While I do not forcibly agree with all the cases enumerated here, I must admit that this new way of putting things feels quite compelling. There seems to indeed be a need for quick and efficient methods, at least for a few targeted applications. Your comment made me reconsider this specific complaint I had.

---

> ### Author Response · Authors · 2019-11-11
> **Response 8**
>
>
> ##########################################################################
> Computing saliency maps on a subset of the dataset once a few epochs already gives a good idea of what the network is doing. In any case, since these saliancy maps are intended for human use, I am not convinced about the importance of computing them for each training example at each epoch. Overall, in my opinion, being efficient at generating saliency maps is a nice to have, but not much more.
> ##########################################################################
>
> We added a full discussion on efficiency. We state that visualizing a full batch every 20 iterations adds at least 2.5% overhead. This jumps to 37.5% for SmoothGrad with 15 iterations (per Hooker et al.). Granted 2.5% is not a killer, but the accuracy per Hooker et al. is highly suspect. We are forced to use SmoothGrad in order to obtain gradient results on par with the method presented here. If we run our method every iteration, the extra overhead is between 0.2 and 1%.

---

> ### Author Response · Authors · 2019-11-11
> **Misc Reponses**
>
>
> ##########################################################################
> Some general comments:
>
> - In Sec. 2.2, the last "con" seems a bit out of place. This could be applied to pretty much anything.
> ##########################################################################
>
> We got similar feedback from some of the peers we shared our paper with. We will remove it.
>
> ##########################################################################
> - Sec. 2.3 is interesting, but the explanations are convoluted. In essence, what should be said is 1) value encompasses the importance of a pixel, 2) saturation presents the max-min of the distribution and 3) hue shows the position in the network.
> ##########################################################################
>
> We will try and massage this part a little more.
>
> ##########################################################################
> Also, superimposing these HSV maps over gray scale version of the image like is Fig.2 is difficult to analyze because the "gray" of the image can be confused with the saturation channel.
> ##########################################################################
>
> We got similar feedback from peers we shared the paper with. The non-blended versions were in the appendix, but we will swap them into the main body figure. The alpha blended versions, if anyone wants them, will still be in the appendix. Updated figure can be seen below.
>
> ##########################################################################
> - On p.2, "this is proceeded" -> "this is preceded"?
> ##########################################################################
>
> Fixed

---

### Official Review · AnonReviewer1 · 2019-10-23
**Official Blind Review #1**

**Rating:** 6

**Review:**

I Summary

The authors present a method that computes a saliency map after each scale block of a CNN and combines them according to the weights of the prior layers in a final saliency map. The paper gives two main contributions: SMOE, which captures the informativeness of the corresponding layers of the scale block and LOVI, a heatmap based on HSV, giving information of the region of interest according to the corresponding scale blocks. The method is interesting as it does not require multiple backward passes such as other gradient-based method and thus prove to be more time-efficient.

II Comments
1. Content
Overall the paper is very interesting, but it is not always clear what the contributions are.The authors refer to "scale" in CNNs, this could be described a little as to explain what are scale blocks in a CNN, before introducing the terminology.
The proposed method "LOVI" is promising and I like the use of HSV for the different components. However, it is hard to read, especially when alpha-blended with a grayscale version of the original image.
- The introduction doesn't clearly state what are the contributions of the method, an example of possible applications would be appreciated (the following about robotic for example but with more details)
- 3.1 I really enjoy the fact that different datasets are used for their different properties (foreground, background, sparsity), this is a nice touch
Figure 4 results are a little underwhelming, SMOE's scores are very close to the other methods
- 3.2 The authors refer to Smoothgrad squared method, it is indeed a good process to refine saliency maps, however why it is used could be detailed, just as the parameters chosen for its implementation.
- 4. The authors claim their implementation is "approximately x times faster" but there is no quantitative proof of it, which seems to be one of the selling-point of the paper (or at least one of the best results)

2. Writing
Those typos do not impact the review score, I hope it can help the authors to gain more clarity in their writing.
- Abstract: "it is also quantitatively similar or better in accuracy" -> shouldn't it be "and" instead of "or"?
- Intro
"a gradient saliency map by trying to back-propagate a signal from one end of the network and project it onto the image plane" not well articulated, "by trying to back-propagate" -> "by back-propagating" (the claims seems weak otherwise)
"is running a fishing expedition post hoc" ;)
"An XAI tool which is too expensive will slow down training" Computationally expensive?
- 2.1
"The resemblance to conditional entropy should be apparent" -> is apparent
"we might say it is" -> it is (don't weaken your claims)
"we simple apply" -> we simply
- 3.1
"if the results appeared terrible" -> terrible is too strong/not adapted. What is a bad result and why?
after eq 7 " the second method is in information" -> an information
 -3.2
"which locations are most salient correctly" -> word missing?
- Figure 5: "Higher accuracy values are better results for it" -> yield better results

The phrasing with"one" as in "if one would like to etc" is used a lot through the paper, it can be a little redundant at times.

III Conclusion
The paper is interesting, however, one of the major contributions seems to be the speed of the method but no quantitative results have been reported. I would really appreciate seeing some experiments over it. Overall the results of the obtained maps are not very convincing compared to existing methods. I believe the writing of the paper could be wrapped around the speed of the method and in which context it would be important (robotic, medical?).  The conclusion and discussion are short and could be filled a little more (some captions could be shortened in order to give more space).

Edit: The authors have answered most of my concerns and I am happy to re-evaluate my score to weak accept.

**Experience Assessment:**

I have read many papers in this area.

**Review Assessment: Checking Correctness Of Derivations And Theory:**

I assessed the sensibility of the derivations and theory.

**Review Assessment: Checking Correctness Of Experiments:**

I assessed the sensibility of the experiments.

**Review Assessment: Thoroughness In Paper Reading:**

I read the paper at least twice and used my best judgement in assessing the paper.

---

> ### Author Response · Authors · 2019-11-09
> **Thank you <<<Read Me First>>>**
>
> We would like to thank the reviewer for their helpful suggestions and comments. We are posting our responses first and will add the revised paper soon. Traditionally, these come at the same time. However, given the spirit of this open review process, it seems prudent to allow time for discussion. Hopefully we are not misinterpreting the intent of the conference organizers.
>
> Much of our response as well as edits to the existing manuscript involve questions regarding efficiency. Indeed, it’s in the title. Given the comments, it is obvious that we did not give it the proper treatment. Our paper will add usage examples to the introduction and a breakdown of both time and memory efficiency in the discussion.

---

> ### Author Response · Authors · 2019-11-09
> **Section 2.1 Comments**
>
>
> ##########################################################################
> II Comments
>
> 1. Content Overall the paper is very interesting, but it is not always clear what the contributions are.
> ##########################################################################
>
> Hopefully this seems rectified by our responses that follow.
>
> ##########################################################################
> The authors refer to "scale" in CNNs, this could be described a little as to explain what are scale blocks in a CNN, before introducing the terminology.
> ##########################################################################
>
> Fixed.
>
> ##########################################################################
> The proposed method "LOVI" is promising and I like the use of HSV for the different components.
> ##########################################################################
>
> We would like the reviewers to consider this part of the novelty of our work. Admittedly, this is a machine learning conference and not a conference on computational visualization. However, one key aspect of explainable AI is figuring out how to visualize things in very high dimensions.
>
> ##########################################################################
> However, it is hard to read, especially when alpha-blended with a grayscale version of the original image.
> ##########################################################################
>
> We got similar feedback from peers we shared the paper with. The non-blended versions were in the appendix, but we will swap them into the main body figure. The alpha blended versions, if anyone wants them, will still be in the appendix. Updated figure can be seen below.

---

> ### Author Response · Authors · 2019-11-09
> **Section 2.2 Comments**
>
>
> ##########################################################################
> - The introduction doesn't clearly state what are the contributions of the method, an example of possible applications would be appreciated (the following about robotic for example but with more details)
> ##########################################################################
>
> We are adding examples and details. The saliency methods where we have seen quantitative results of efficacy have at least one of three components which create efficiency related issues we discuss below. These components are:
>
> (1)	Usage of gradient data.
> (2)	Usage of multiple passes through the network.
> (3)	Usage of saliency network training.
>
> Below are examples where our method is efficient enough to work well, but other methods are not. The discussion will show a detailed breakdown of our method’s efficiency (see further below). These are divided by efficiency type. The introduction will contain this information, but organized in a standard way, not just a long list.
>
> 1.	Examples of efficiency related to time complexity: This affects methods which use gradient data and multiple network passes. Some saliency network training models may be affected, but we know of some which are not. Examples are:
>     a. If a person is using real-time vision applications where embedded hardware is sometimes barely fast enough. Doing an extra backward pass in the field can lead to a dropped frame each time it is computed.
>         i. Frame drop can also affect data recording.
>     b. In robotics, engineers typical like to see real-time feedback while the robot is running. Per (1.a.), methods that require full backward gradients are less feasible due to irritating frame drop.
>         i. Answering questions like “Why did the robot run into the wall?”, “What was it looking at?”, or “Can I quickly tweak a parameter and get it to behave?” are interactive and iterative in practice, so real-time diagnostics and feedback provided by efficient saliency maps are crucial.
>     c.	Time efficiency is important in cloud computing where GPU usage translates into dollars (or carbon).
>     d. Another example is DNNs that use saliency as part of their pipeline. Hypothetically, this means much faster training.
> 2.	Examples of efficiency related to memory complexity: This affects methods which use gradients and may affect any method using a trained saliency network. This is a much more difficult issue to overcome and gradient methods in particular would be untenable in these scenarios since the information required for a backward pass is rather large. These are examples using deployed, pre-trained networks.
>     a. Processing of very large satellite images may preclude the storage of information required to compute gradients.
>     b. Deployed industrial inspection systems frequently have lower cost GPUs with less memory. These are sufficient for running a model but not training one. When these systems fail in production, engineers want to see what is going on. It’s nice if diagnostic information is presented online.
>     c.	Many embedded systems might not only lack sufficient memory in general, but may not even have to ability to store or compute gradient data.
>     d. Methods utilizing a pre-trained saliency network may work in some of these situations so long as they are not too large. However, the addition of even a handful of extra layers may rule out their usage.
> 3.	Issues related to logistical efficiency. This would primarily affect methods which must be pre-trained. It comes from the synchrony of maintaining a trained model in the field and its complementary trained saliency network. It introduces a source of human error. Extra effort must be used to make sure both models are kept in sync.
>
> In summary, we have created the only explainable AI saliency method which will work unconditionally in all these kinds of conditions and has demonstrable and quantitatively good results.

---

> ### Author Response · Authors · 2019-11-09
> **Section 2.3 Comments**
>
>
> ##########################################################################
> - 3.1 I really enjoy the fact that different datasets are used for their different properties (foreground, background, sparsity), this is a nice touch Figure 4 results are a little underwhelming, SMOE's scores are very close to the other methods
> ##########################################################################
>
> (1)	The layer 1 scores appear closer than they are. We will mention that the layer 1 Difference score for the second best metric, Standard Deviation is 0.0031 . This means its performance is almost the same as a random saliency map. SMOE Scale is 0.1051. Only Log Normal Mean and Shannon Entropy score near it. However, those two do not do very well overall.
> (2)	We will add a table to show that a difference of 0.102 between Standard Deviation and SMOE Scale is rather large since for layers 3 through 5, there is never a difference greater than 0.0625 between any of tested methods.
> (3)	We are adding a graphic to show that Standard Deviation has a tendency to flood on the layer 1 maps and fail to catch low level features one would expect to be important. This suggests why Standard Deviation appears to fail on this layer.
> (4)	Layer 1 is important since it carries the finest spatial details for the combined saliency map.

---

> ### Author Response · Authors · 2019-11-09
> **Section 2.4 Comments**
>
>
> ##########################################################################
> - 3.2 The authors refer to Smoothgrad squared method, it is indeed a good process to refine saliency maps, however why it is used could be detailed, just as the parameters chosen for its implementation.
> ##########################################################################
>
> (1)	The original ROAR/KARR paper of Hooker et al.  tested 12 different variations and methods. The three SmoothGrad methods appear to be the strongest three from their results.
> (2)	Non SmoothGrad results are in the appendix. However, we will integrate these into Table 2. As Hooker et al. notes, non-iterative gradient methods perform very poorly. Only SmoothGrad and VarGrad yield reasonable results. We will include more text about this.

---

> ### Author Response · Authors · 2019-11-09
> **Section 2.5 Comments**
>
>
> ##########################################################################
> - 4. The authors claim their implementation is "approximately x times faster" but there is no quantitative proof of it, which seems to be one of the selling-point of the paper (or at least one of the best results)
> ##########################################################################
>
> We sincerely apologize for this lapse. Back of the envelope calculations really are not acceptable for this venue. We will include a full discussion section to detail time complexity, memory complexity and operation counts. Operations from our method come from three sources. The first is the computation of statistics on tensors in five layers. The second is the normalization of each 2D saliency map. Third we account for the cost of combing the saliency maps.
>
> Ops for our solution ranges from 1.1 x 10^7 to 3.9 x 10^7 FLOPs (using terminology from He 2015 [the original ResNet paper]) for a ResNet-50. The network itself has 3.8 x 10^9 FLOPs in total. The range in our method comes from how one counts Log and Error Function operations which are operationally expensive compared to more standard ops.  How they work is apparently a Nvidia trade secret, so we estimate the worst case from available software instantiations. Most of the work comes from the initial computation of statistics over activation tensors. This ranges from 9.2 x 10^6 to 3.7 x 10^7 FLOPs. In total, this gives our model an overhead of 0.3% to 1.0% relative to the network itself. All full pass methods have a nominal overhead of 100%.
>
> Compared to any method which requires a full pass, such as gradient methods, our solution is nominally between 97x and 344x faster for non-SmoothGrad techniques, which according to Hooker et al. performs poorly on ROAR/KAR scores. We are between 1456x and 5181x faster than a 15-iteration SmoothGrad implementation that yields the results in figure 5. 15 iterations as well as other parameters was chosen by Hooker et al. who describes this selection in more detail.
>
> The memory footprint of our model is minuscule. Computation over tensors can be done inline which leaves the largest storage demand being the retention of 2D saliency maps. This is increased slightly by needing to store one extra 112x112 image during bilinear up-sampling. Peak memory overhead related to data storage is about 117 kilobytes per 224x224 input image.
>
> We will attach detailed tables to the Appendix showing our work.

---

> ### Author Response · Authors · 2019-11-09
> **Section 3 Comments**
>
>
> ##########################################################################
> 2. Writing Those typos do not impact the review score, I hope it can help the authors to gain more clarity in their writing.
> - Abstract: "it is also quantitatively similar or better in accuracy" -> shouldn't it be "and" instead of "or"?
> ##########################################################################
>
> Changed.
>
> ##########################################################################
> - Intro "a gradient saliency map by trying to back-propagate a signal from one end of the network and project it onto the image plane" not well articulated,
> ##########################################################################
>
> Fixed.
>
> ##########################################################################
> "by trying to back-propagate" -> "by back-propagating" (the claims seems weak otherwise)
> ##########################################################################
>
> Fixed.
>
> ##########################################################################
> "is running a fishing expedition post hoc" ;)
> ##########################################################################
>
> Fixed.
>
> ##########################################################################
> "An XAI tool which is too expensive will slow down training" Computationally expensive?
> ##########################################################################
>
> Fixed.
>
> ##########################################################################
> - 2.1 "The resemblance to conditional entropy should be apparent" -> is apparent
> ##########################################################################
>
> Fixed.
>
> ##########################################################################
> "we might say it is" -> it is (don't weaken your claims)
> ##########################################################################
>
> We got similar feedback from peers we shared the paper with. Fixed.
>
> ##########################################################################
> "we simple apply" -> we simply
> ##########################################################################
>
> Fixed.
>
> ##########################################################################
> - 3.1 "if the results appeared terrible" -> terrible is too strong/not adapted. What is a bad result and why?
> ##########################################################################
>
> Terrible was a terrible word to use. Author 1 thought it seemed to sound awkward, but left it in anyways, bad choice.
>
> ##########################################################################
> after eq 7 " the second method is in information" -> an information
> ##########################################################################
>
> Fixed.
>
> ##########################################################################
> -3.2 "which locations are most salient correctly" -> word missing?
> ##########################################################################
>
> Fixed.
>
> ##########################################################################
> - Figure 5: "Higher accuracy values are better results for it" -> yield better results
> ##########################################################################
>
> Fixed.
>
> ##########################################################################
> The phrasing with"one" as in "if one would like to etc" is used a lot through the paper, it can be a little redundant at times.
> ##########################################################################
>
> Author 1 has annoyed reviewers on more than one occasion with the over usage of a word in this exact same way. Author 1 needs to be more vigilant in the future.

---

### Author Response · Authors · 2019-11-14
**General Comments and Thank You**

We would like to thank all the reviewers for their helpful comments and suggestions. It is appropriate to treat peer review as a focus group and an opportunity to learn from other experts we may not interact with on any regular basis. We saw that there were two common concerns from the reviewers. The first was that it was unclear how important efficiency is for our application. The second involved questions about how much better SMOE Scale is than trivial statistics. We have made changes to the manuscript to address these concerns. Hopefully the changes as well as our individual responses will be satisfactory to the reviewers.

We would like to direct the reviewers to a few other contributions of this manuscript that may have gone unnoticed.
(1)	The LOVI method for visualizing high dimensional activation maps is novel on its own and will probably be helpful for applications other than what we have presented.
(2)	The fact that the weighted sum of trivial statistics over a few layers does so well compared with Guided Backprop seemed rather startling to us. It raises questions in our mind about the way we assume information moves through a CNN.
(3)	If you are still not convinced by SMOE Scale, at least you now know which simple statistics to use that give the best result. Even standard deviation does better than Smooth Guided Backprop. It’s not as reliable as SMOE Scale, but you still get the three order of magnitude speedup. We have saved you several months of computation time, running ROAR/KAR, to figure this out.
(4)	We have illuminated the idea of saliency map order equivalence (SMOE). By keeping this in mind, a person can construct more optimal solutions in the future. It’s a simple trick, but also a very helpful one. There is no point in computing terms which do not change the order of a saliency map.

A note to the chair:
If we are correct in how useful our technique will be to the mobile, robotics, industrial and embedded device community, this work should garner plenty of citations. We appear to have the only demonstrably accurate and resource feasible XAI saliency map solution in many cases.

We have added a few pages to our manuscript to reply to the reviewers. We have stayed within the 10-page hard limit. Looking at the author guidelines it is unclear if the desired eight-page limit applies also to the post review revised manuscript. Generally speaking, it is common for conferences to give a few extra pages for this purpose. Talking with some of our peers, they had the same understanding of the authors guidelines as well. We would like to ask the conference to make this clearer next year.

---

### Public Comment · ~Suraj_Srinivas1 · 2020-03-03
**Concurrent work - FullGrad**

Dear authors,

I just wanted to point out for reference our concurrent work similar to this paper, which is regarding the full-gradient decomposition (https://arxiv.org/abs/1905.00780), published in NeurIPS 2019. The code is available online as well (https://github.com/idiap/fullgrad-saliency).

Cheers!

Regards,
Suraj Srinivas

---

### Decision · Program_Chairs · 2019-12-19

**Decision:**

Reject

**Comment:**

The paper presents an efficient approach to computer saliency measures by exploiting saliency map order equivalence (SMOE), and visualization of individual layer contribution by a layer ordered visualization of information.

The authors did a good job at addressing most issues raised in the reviews. In the end, two major concerns remained not fully addressed: one is the motivation of efficiency, and the other is how much better SMOE is compared with existing statistics. I think these two issue also determines how significance the work is.

After discussion, we agree that while the revised draft pans out to be a much more improved one, the work itself is nothing groundbreaking. Given many other excellent papers on related topics, the paper cannot make the cut for ICLR.